# Carotenoid assembly regulates quinone diffusion and the *Roseiflexus castenholzii* reaction center-light harvesting complex architecture

Jiyu Xin[1†], Yang Shi[2†], Xin Zhang[1,3†], Xinyi Yuan[1,3], Yueyong Xin[3], Huimin He[3], Jiejie Shen[1], Robert E Blankenship[4], Xiaoling Xu[1,3]*

[1]Department of Biochemistry and Molecular Biology, School of Basic Medical Sciences and The Affiliated Hospital of Hangzhou Normal University, Hangzhou, China; [2]Liangzhu Laboratory, MOE Frontier Science Center for Brain Science and Brain-machine Integration, State Key Laboratory of Brain-machine Intelligence & Department of Neurobiology and Department of Pathology of the First Affiliated Hospital, Zhejiang University School of Medicine, Zhejiang University, Hangzhou, China; [3]Photosynthesis Research Center, College of Life and Environmental Sciences, Hangzhou Normal University, Hangzhou, China; [4]Departments of Biology and Chemistry, Washington University in St. Louis, St. Louis, United States

**\*For correspondence:**
xuxl@hznu.edu.cn

[†]These authors contributed equally to this work

**Competing interest:** The authors declare that no competing interests exist.

**Abstract** Carotenoid (Car) pigments perform central roles in photosynthesis-related light harvesting (LH), photoprotection, and assembly of functional pigment-protein complexes. However, the relationships between Car depletion in the LH, assembly of the prokaryotic reaction center (RC)-LH complex, and quinone exchange are not fully understood. Here, we analyzed native RC-LH (nRC-LH) and Car-depleted RC-LH (dRC-LH) complexes in *Roseiflexus castenholzii*, a chlorosome-less filamentous anoxygenic phototroph that forms the deepest branch of photosynthetic bacteria. Newly identified exterior Cars functioned with the bacteriochlorophyll B800 to block the proposed quinone channel between LHαβ subunits in the nRC-LH, forming a sealed LH ring that was disrupted by transmembrane helices from cytochrome *c* and subunit X to allow quinone shuttling. dRC-LH lacked subunit X, leading to an exposed LH ring with a larger opening, which together accelerated the quinone exchange rate. We also assigned amino acid sequences of subunit X and two hypothetical proteins Y and Z that functioned in forming the quinone channel and stabilizing the RC-LH interactions. This study reveals the structural basis by which Cars assembly regulates the architecture and quinone exchange of bacterial RC-LH complexes. These findings mark an important step forward in understanding the evolution and diversity of prokaryotic photosynthetic apparatus.

## Editor's evaluation

This is a valuable analysis of the structure of *Roseiflexus castenholzii* native and carotenoid-depleted light harvesting complexes. The authors have investigated the relationship between Carotenoid pigment depletion in the photosynthesis-related light harvesting complex, the assembly of the prokaryotic reaction center LH complex, and quinone exchange in *Roseiflexus castenholzii*, a chlorosome-less filamentous anoxygenic phototroph that forms the deepest branch of photosynthetic bacteria. The evidence supporting the claims is solid, with application of rigorous biochemical and biophysical techniques, including cryo-electron microscopy of the purified RC-LH complexes

with or depleted of carotenoids. This study will be of interest to biologists working on the evolution and diversity of prokaryotic photosynthetic apparatus.

## Introduction

Carotenoids (Cars) are natural pigments that play important roles in light harvesting (LH), photo-protection, and assembly of the functional pigment-protein complexes required for photosynthesis. Specifically, Cars capture blue-green light (450–550 nm) and transfer it to chlorophyll or bacteriochlorophyll ((B)Chl) in the LH antenna. The excited energy is then transferred to the RC for primary photochemical reactions. In anoxygenic photosynthetic bacteria (PSB), Car-BChl interactions are essential for assembling the functional LH complexes (*Davidson and Cogdell, 1981*; *Hashimoto et al., 2016*; *Lang and Hunter, 1994*; *Walz and Ghosh, 1997*). The well-studied purple bacterium *Rhodobacter* (*Rba.*) *sphaeroides* contains a closed LH2 ring comprising nine αβ-polypeptides; each LHαβ non-covalently binds three BChls (two B850s and one B800) and one Car (*Qian et al., 2021b*). The Car-less strains of *Rba. sphaeroides* are unable to assemble an LH2 complex, indicating that Car-BChl interactions are essential for the maintenance of LH2 structural stability (*Lang et al., 1995*). In the LH1 ring of *Rba. sphaeroides*, a combination of two Car groups forms a tightly sealed, impenetrable fence-like structure that blocks the proposed quinone channel of the closed ring (*Olsen et al., 2017*; *Qian et al., 2021c*). However, there are fewer Cars in most LH1 structures, so in *Thermochromatium* (*Tch.*) *tepidum* and *Rhodospirillum* (*Rsp.*) *rubrum* for example, there are small gaps that allow quinones to shuttle cross the ring (*Niwa et al., 2014*; *Qian et al., 2021a*; *Yu et al., 2018b*). A point mutation in LHα (W24F) dramatically reduces the amounts of LH1-bound Car. However, in the *pufX* knockout strain of *Rba. sphaeroides*, which possesses a closed LH1 ring composed of 17 LHαβs, the same mutation promotes photosynthetic growth (*Cao et al., 2022*; *McGlynn et al., 1994*; *Olsen et al., 2017*). These observations indicate a correlation between the number of LH1-bound Cars and the architecture and photochemical functions of the RC-LH1. This phenomenon could be further studied using structural information about Car-depleted RC-LH (dRC-LH), but no such data have yet been reported.

*Roseiflexus* (*R.*) *castenholzii* is a chlorosome-less filamentous anoxygenic photosynthetic bacterium (*Hanada et al., 2002*). It contains only one LH, which forms an unusual RC-LH complex. This complex structurally resembles RC-LH1 but has similar spectroscopic characteristics that are similar to the peripheral LH2 of purple bacteria (*Collins et al., 2010*; *Collins et al., 2009*). We previously reported the cryo-electron microscopy (EM) structure of *R. castenholzii* RC-LH at 4.1 Å resolution. It revealed an RC composed of L, M, and cytochrome (cyt) *c* subunits surrounded by an opened elliptical LH ring of 15 LHαβs, with the tetraheme binding domain of cyt *c* protruding on the periplasmic side. The RC is compositionally larger in purple bacteria than in *R. castenholzii*, in which it does not contain an H subunit (*Pugh et al., 1998*; *Qian et al., 2005*; *Yamada et al., 2005*). However, it does contain a unique cyt *c* transmembrane (c-TM) helix and the newly identified subunit X, both of which flank the gap of the LH ring to form a novel quinone shuttling channel (*Xin et al., 2018*). Notably, the amino acid sequences of subunit X and TM7, a TM helix separated from the RC-L and RC-M subunits are unassigned. Pigment analyses have revealed a 2:3 Car:BChl molar ratio of *R. castenholzii* RC-LH (*Collins et al., 2009*). However, the cryo-EM structure resolved only one keto-γ-carotene (KγC) molecule spanning the interface of each LHαβ, coordinating two B880s and one additional B800 at the periplasmic and the cytoplasmic side, respectively. The lack of a clear cryo-EM density map leaves uncertainty about the presence of additional LH ring-bound Cars, the roles of which are unknown in maintaining the architecture and photochemical functions of the *R. castenholzii* RC-LH.

We here determined cryo-EM structures of native RC-LH (nRC-LH) complexes purified from *R. castenholzii* cells grown under high (180 µmol m$^{-2}$ s$^{-1}$), medium (32 µmol m$^{-2}$ s$^{-1}$), and low (2 µmol m$^{-2}$ s$^{-1}$) illuminations at 2.8 Å, 3.1 Å, and 2.9 Å resolutions, respectively. All three structures shared the same architecture, indicating that the Car composition and assembly are not affected by light intensities. From these high-resolution structures, we identified 14 additional KγC molecules in the exterior of the LH ring (KγC$_{ext}$). In combination with the B800 on the cytoplasmic side, the newly identified KγC$_{ext}$ molecules blocked the proposed quinone channel between LHαβ subunits, forming a sealed LH ring conformation. We also assigned the full amino acid sequences of subunit X, TM7, and an additional TM helix that were derived from hypothetical proteins Y and Z, respectively, and demonstrated their roles in forming the quinone channel and stabilizing the RC-LH interactions. To investigate the

**eLife digest** Photosynthesis is a biological process that converts energy from sunlight into a form of chemical energy that supports almost all life on Earth. Over the course of evolution, photosynthesis has gone from being only performed by bacteria to appearing in algae and green plants. While this has given rise to a range of different machineries for photosynthesis, the process always begins the same way: with a structure called the reaction center-light harvesting (RC-LH) complex.

Two pigments in the light-harvesting (LH) region – known as chlorophyll and carotenoids – absorb light energy and transfer it to another part of the complex known as the quinone-type reaction center (RC). This results in the release of electrons that interact with a molecule called quinone converting it to hydroquinone. The electron-bound hydroquinone then shuttles to other locations in the cell where it initiates further steps that ultimately synthesize forms of chemical energy that can power essential cellular processes.

In photosynthetic bacteria, hydroquinone must first pass through a ring structure in the light harvesting region in order to leave the reaction center. Previous studies suggest that carotenoids influence the architecture of this ring, but it remains unclear how this may affect the ability of hydro-quinone to move out of the RC-LH complex.

To investigate, Xin, Shi, Zhang et al. used a technique called cryo-electron microscopy to study the three-dimensional structure of RC-LH complexes in one of the first bacterial species to employ photosynthesis, *Roseiflexus castenholzii*. The experiments found that fully assembled complexes bind two groups of carotenoids: one nestled in the interior of the LH ring and the other on the exterior.

The exterior carotenoids work together with bacteriochlorophyll molecules to form a closed ring that blocks hydroquinone from leaving the RC-LH complex. To allow hydroquinone to leave, two groups of regulatory proteins, including a cytochrome and subunit X, then disrupt the structure of the ring to 'open' it up.

These findings broaden our knowledge of the molecules involved in photosynthesis. A better understanding of this process may aid the development of solar panels and other devices that use RC-LH complexes rather than silicon or other inorganic materials to convert energy from sunlight into electricity.

role of Cars in the assembly of RC-LH, *R. castenholzii* cells were treated with Car biosynthesis inhibitor diphenylamine (DPA) to produce a dRC-LH; a 3.1 Å resolution cryo-EM structure of this complex resolved five KγC molecules bound in the interior of the LH ring (KγC$_{int}$). The absence of subunit X and exterior KγC (KγC$_{ext}$) molecules in the dRC-LH produced an LH ring with exposed LHαβ interface and a larger opening than that of nRC-LH. This conformation accelerated the in vitro quinone/quinol exchange rate of menaquinone-4, an analog of the native menaquinone-11, but did not affect the Car-to-BChl energy transfer efficiency of dRC-LH. This study thus revealed a previously unrecognized structural basis by which Car assembly regulates the architecture and quinone/quinol exchange rate of the bacterial RC-LH complex. These findings further our understanding of diversity and molecular evolution in the prokaryotic photosynthetic apparatus.

## Results

### Identification of KγC$_{ext}$ in the nRC-LH complex

To investigate the LH-bound Car numbers and its correlation with the light intensities, we anaerobically cultured *R. castenholzii* cells under the light intensity (32 μmol m$^{-2}$ s$^{-1}$) used for obtaining the reported 4.1 Å RC-LH structure (*Xin et al., 2018*), and also a high and a low light intensity at 180 μmol m$^{-2}$ s$^{-1}$ and 2 μmol m$^{-2}$ s$^{-1}$, respectively. For easier reading, we labeled these three light intensities as high (180 μmol m$^{-2}$ s$^{-1}$), medium (32 μmol m$^{-2}$ s$^{-1}$), and low (2 μmol m$^{-2}$ s$^{-1}$) illuminations. The cell proliferation rate was much faster under high illumination than that grown under medium and low illuminations, and the cells grown showed a darker reddish-brown color after 120 hr of culturing (*Figure 1—figure supplement 1A and B*). We then isolated and purified nRC-LH complexes from these cells at the stationary growth phase (*Figure 1—figure supplement 1C*, *Table 1*). Ultraviolet (UV)-visible-near infrared (NIR) spectrophotometry of the isolated nRC-LH complexes showed

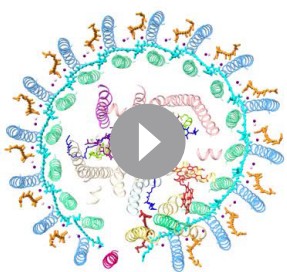

**Video 1.** Top view of the conformational changes between native reaction center-light harvesting (RC-LH) (nRC-LH) and carotenoid-depleted RC-LH (dRC-LH) complexes from *R. castenholzii*. The color scheme is same as *Figures 1 and 4*.

https://elifesciences.org/articles/88951/figures#video1

typical $Q_y$ bands at 800 nm (B800) and 880 nm (B880) and a $Q_x$ band at 594 nm, which corresponded to LH-bound BChls. Notably, Car-associated absorption peaks were detected at 457 nm, 482 nm, and 519 nm (*Figure 1—figure supplement 1D*). The nRC-LH complexes purified from cells under high, medium, and low illuminations showed the same Car absorption spectrum (*Figure 1—figure supplement 1E*), indicating the pigments content was not affected by light intensities. These nRC-LH complexes were then imaged via cryo-EM, respectively (*Figure 1—figure supplements 2 and 3*). Using single particle analysis, the nRC-LH structures obtained from the high, medium, and low illumination cultured cells were resolved at an overall resolution of 2.8 Å, 3.1 Å, and 2.9 Å, respectively (*Figure 1—figure supplements 2 and 4*, *Table 2*). Superposition of the high illumination model with that of medium and low illumination gave root mean square deviation of 1.753 Å and 1.765 Å, respectively, indicating these three structures share the same architecture, and light intensities did not affect the conformation of the nRC-LH structures.

The 15 LHαβ heterodimers formed an opened elliptical ring surrounding the RC, which contained L, M, and cyt c subunits; the long and short axes were 112 Å and 103 Å, respectively, and a tetraheme binding domain of cyt *c* protruded into the periplasmic space (*Figure 1A and B*). Similar as most purple bacteria, the RC contained a photo-reactive special pair of BChls, one accessory BChl, three bacteriopheophytins (BPheos), two MQ-11 ($MQ_A$ and $MQ_B$) and a newly identified $MQc$, and an iron atom to mediate the charge separation and subsequent electron transfer (*Figure 1C*). Each LHαβ non-covalently bound two B880s and one B800 BChl on the periplasmic and cytoplasmic sides (*Figures 1C and 2A*). In particular, the LH ring bound 15 $KγC_{int}$, 14 $KγC_{ext}$ Cars, and an additional KγC that inserted between the LHαβ1 and c-TM in all three structures (*Figure 1B–D*, *Figure 1—figure supplement 5*, *Video 1*), indicating both Car compositions and assembly in the nRC-LH were not affected by light intensities. The low-pass filtered cryo-EM map of nRC-LH minus that of the reported 4.1 Å model showed apparent density differences for the $KγC_{ext}$ (*Figure 1—figure supplement 6*), indicating the $KγC_{ext}$ molecules were not resolved due to lack of clear EM densities in the 4.1 Å model. Given the similarities between these three nRC-LH structures, we use the 2.8 Å model for following analyses of the nRC-LH structure.

**Table 1.** Peptide mass fingerprinting (PMF) analysis of the *R. castenholzii* in reaction center-light harvesting (RC-LH) that are separated by blue-native PAGE.

| Subunit | Accession | Description | Score | Coverage |
|---|---|---|---|---|
| cyt *c* | BAC76415.1 | Cytochrome subunit of photosynthetic reaction center (*R. castenholzii*) | 131.79 | 40.94% |
| L and M | BAC76414.1 | Precursor for L and M subunits of photosynthetic reaction center (*R. castenholzii*) | 195.79 | 28.39% |
| LHα subunit | BAC76413.1 | Alpha subunit of light harvesting 1 (*R. castenholzii*) | 97.78 | 100% |
| LHβ subunit | BAC76412.1 | Beta subunit of light harvesting 1 (*R. castenholzii*) | 49.11 | 100% |
| Z subunit | WP_041331144.1 | Hypothetical protein (*R. castenholzii*) | | 19.05% |

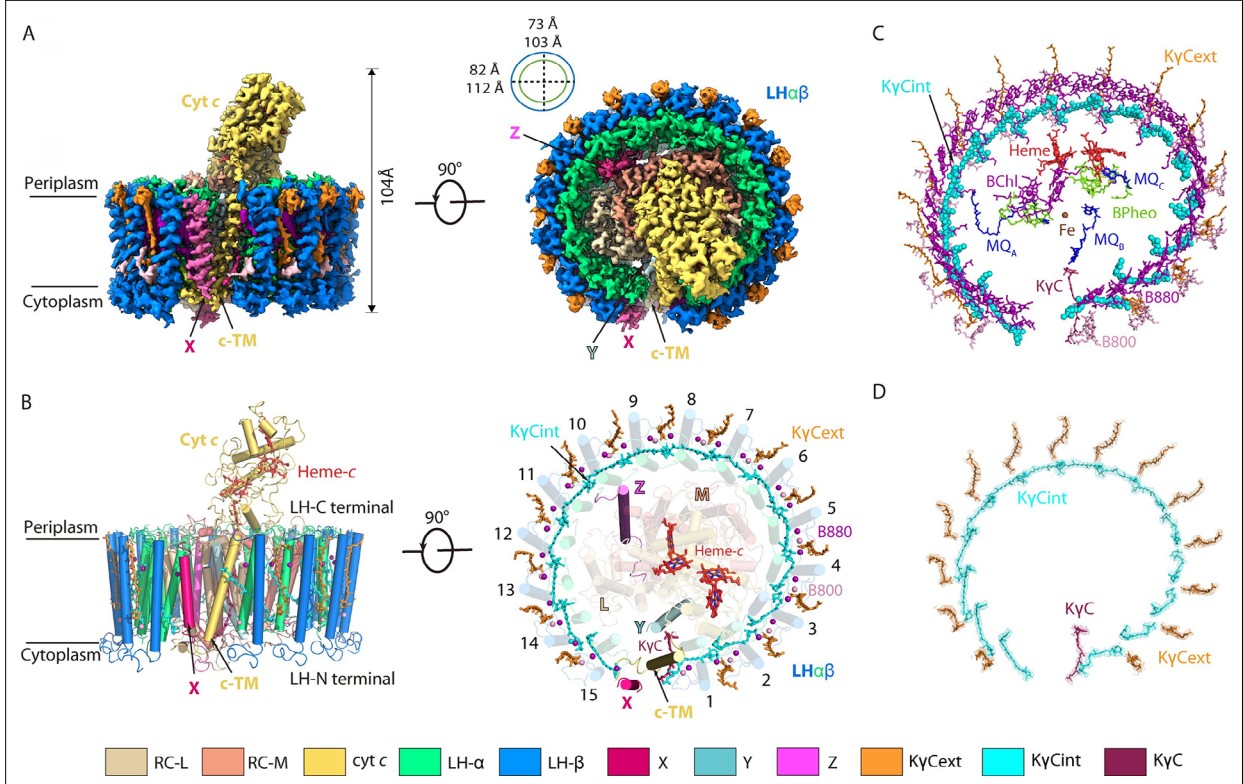

**Figure 1.** Overall structure of the native reaction center (RC)-light harvesting (LH) complex from *R. castenholzii*. (**A**) A cryo-electron microscopy (cryo-EM) map of the native RC-LH (nRC-LH) complex is shown from the side (left panel) and the bottom (right panel). The dimensions of the RC-LH complex and LH ring are represented. The positions of subunit X, proteins Y and Z, and the cytochrome (cyt) *c* transmembrane (c-TM) domain are labeled. (**B**) Side and top views of the nRC-LH complex are presented in cartoon form. LH subunits are numbered clockwise from the gap formed by subunit X and c-TM. Heme-*c* (red) and keto-γ-carotene (KγC) molecules (orange, cyan, ruby) are shown in stick forms; Mg atoms of the bacteriochlorophylls B800 (pink) and B880 (purple) are shown as spheres. (**C**) The cofactors bound in the nRC-LH complex. All cofactors are shown in stick forms except for the interior KγC (KγC$_{int}$) in LH, the iron bound in the RC are shown as spheres. (**D**) The structural models of the KγC$_{int}$, exterior KγC (KγC$_{ext}$), and KγC in the nRC-LH complex are fitted in the EM density map. The color scheme: lime green, α-polypeptides; marine, β-polypeptides; yellow-orange, cyt *c*; wheat, L subunit; salmon, M subunit; pale cyan, protein Y; hot pink, subunit X; light magenta, protein Z; cyan, KγC$_{int}$; orange, KγC$_{ext}$; ruby, KγC; purple, B880; pink, B800; tv-red, heme-*c*; chartreuse, bacteriopheophytins (BPheos); blue, menaquinone-11 (MQ); brown, iron.

The online version of this article includes the following source data and figure supplement(s) for figure 1:

**Figure supplement 1.** Purification and verification of the native reaction center-light harvesting (nRC-LH) and carotenoid (Car)-depleted RC-LH (dRC-LH) complexes from *R. castenholzii*.

**Figure supplement 1—source data 1.** Raw figures of the full uncropped blue native PAGE of nRC-LH with and without the relevant bands labelled.

**Figure supplement 1—source data 2.** Raw figures of the full uncropped SDS PAGE of nRC-LH with and without the relevant bands labelled.

**Figure supplement 1—source data 3.** Raw figures of the full uncropped blue native PAGE of dRC-LH with and without the relevant bands labelled.

**Figure supplement 1—source data 4.** Raw figures of the full uncropped SDS PAGE of dRC-LH with and without the relevant bands labelled.

**Figure supplement 2.** Cryo-electron microscopy (cryo-EM) analysis of the native reaction center-light harvesting (nRC-LH) complex from *R. castenholzii*.

**Figure supplement 3.** Cryo-electron microscopy (cryo-EM) analysis of the native reaction center-light harvesting (nRC-LH) complexes purified from *R. castenholzii* grown under medium (32 μmol m$^{-2}$ s$^{-1}$) or low (2 μmol m$^{-2}$ s$^{-1}$) illuminations.

**Figure supplement 4.** Cryo-electron microscopy (cryo-EM) densities and structural models of the reaction center-light harvesting (RC-LH) complex from *R. castenholzii*.

**Figure supplement 5.** The structures of native reaction center-light harvesting (RC-LH) (nRC-LH) complexes obtained from *R. castenholzii* grown under low (2 μmol m$^{-2}$ s$^{-1}$) and medium (32 μmol m$^{-2}$ s$^{-1}$) illuminations.

**Figure supplement 6.** Comparison of the cryo-electron microscopy (cryo-EM) maps of the native reaction center-light harvesting (nRC-LH) and carotenoid-depleted RC-LH (dRC-LH) complexes obtained at medium (32 μmol m$^{-2}$ s$^{-1}$) and high (180 μmol m$^{-2}$ s$^{-1}$) illuminations.

**Table 2.** Cryo-electron microscopy (cryo-EM) data collection, refinement, and validation statistics.

| | Native RC-LH at 180 µmol m⁻² s⁻¹ (nRC-LH) (EMD-34838) (PDB 8HJU) | Carotenoid-depleted RC-LH at 180 µmol m⁻² s⁻¹ (dRC-LH) (EMD-34839) (PDB 8HJV) |
|---|---|---|
| **Data collection and processing** | | |
| Magnification | 64,000 | 81,000 |
| Voltage (kV) | 300 | 300 |
| Electron exposure (e⁻/Å²) | 50 | 49.65 |
| Defocus range (µm) | –1.0 to –2.3 | –1.1 to –1.7 |
| Pixel size (Å) | 1.08 | 0.893 |
| Symmetry imposed | C1 | C1 |
| Initial particle images (no.) | 1,625,156 | 1,081,719 |
| Final particle images (no.) | 372,029 | 84,352 |
| Map resolution (Å) FSC threshold | 2.8 0.143 | 3.1 0.143 |
| **Refinement** | | |
| Initial model used (PDB code) | 5YQ7 | Native RC-LH |
| Model resolution (Å) FSC threshold | 2.8 0.5 | 3.1 0.5 |
| Map sharpening $B$ factor (Å²) | 98 | 120 |
| Model composition Non-hydrogen atoms Protein residues Ligands | 23,917 2330 97 | 22,193 2265 67 |
| $B$ factors (Å²) Protein Ligand | 35.81 32.21 | 52.46 55.37 |
| R.m.s. deviations Bond lengths (Å) Bond angles (°) | 0.011 1.509 | 0.010 1.205 |
| Validation MolProbity score Clashscore Poor rotamers (%) | 2.00 16.11 0.71 | 2.17 18.87 0.84 |
| Ramachandran plot Favored (%) Allowed (%) Disallowed (%) | 95.83 4.17 0.00 | 94.16 5.75 0.09 |

| | Native RC-LH at 2 µmol m⁻² s⁻¹ (EMD-35988) (PDB 8J5O) | Native RC-LH at 32 µmol m⁻² s⁻¹ (EMD-35989) (PDB 8J5P) |
|---|---|---|
| **Data collection and processing** | | |
| Magnification | 81,000 | 81,000 |
| Voltage (kV) | 300 | 300 |
| Electron exposure (e⁻/Å²) | 50 | 50 |
| Defocus range (µm) | –1.2 to –2.0 | –1.2 to –2.0 |
| Pixel size (Å) | 0.893 | 0.893 |
| Symmetry imposed | C1 | C1 |

*Table 2 continued on next page*

*Table 2 continued*

| | Native RC-LH at 2 μmol m⁻² s⁻¹ (EMD-35988) (PDB 8J5O) | Native RC-LH at 32 μmol m⁻² s⁻¹ (EMD-35989) (PDB 8J5P) |
|---|---|---|
| Initial particle images (no.) | 779,594 | 686,082 |
| Final particle images (no.) | 322,595 | 272,617 |
| Map resolution (Å) FSC threshold | 2.9 0.143 | 3.1 0.143 |
| **Refinement** | | |
| Initial model used (PDB code) | 8HJU | 8HJU |
| Model resolution (Å) FSC threshold | 2.8 0.5 | 3.1 0.5 |
| Map sharpening *B* factor (Å²) | 98 | 120 |
| Model composition Non-hydrogen atoms Protein residues Ligands | 23,691 2318 93 | 23,737 2331 92 |
| *B* factors (Å²) Protein Ligand | 27.54 24.73 | 39.01 37.37 |
| R.m.s. deviations Bond lengths (Å) Bond angles (°) | 0.018 1.915 | 0.017 1.850 |
| Validation MolProbity score Clashscore Poor rotamers (%) | 1.97 18.79 0.00 | 1.98 17.69 0.30 |
| Ramachandran plot Favored (%) Allowed (%) Disallowed (%) | 96.79 3.21 0.00 | 96.50 3.46 0.04 |

## Incorporation of KγC$_{ext}$ and B800s together blocked the LHαβ interface

Each LHαβ heterodimer of *R. castenholzii* was stabilized by hydrogen bonding interactions between LHβ-Arg55 and LHα-Asn37 on the periplasmic side, and by LHβ-Gln22 and LHα-Arg4 on the cytoplasmic side (*Figure 2—figure supplement 1A*). These interactions were not resolved in the 4.1 Å model, due to lack of clear cryo-EM densities for the Arg4 and Arg55 residues. The LH-bound B880s and one B800 BChl were coordinated by highly conserved His residues on the periplasmic and cytoplasmic sides (*Figure 2A*, *Figure 2—figure supplement 1*). Incorporation of an additional B800 at the cytoplasmic side of the LH ring resembles the exterior LHh ring of *Gemmatimonas* (*G.*) *phototrophica* RC-dLH, in which the B800s were oriented perpendicular to the plane of the membrane (*Qian et al., 2022*). Superposition of each LHαβ with that of *G. phototrophica* LHh revealed high overlap at the TM helices, with the exception that the B800 porphyrin ring was inclined nearly 60° relative to the *G. phototrophica* LHh-bound B800 (*Figure 2B*). Notably, the B800 conformation was also different from that of B800s bound in *Rba. sphaeroides* LH2 and *R. acidophila* LH3, in which the porphyrin rings were both oriented toward the center of the LH ring (*Figure 2—figure supplement 1C*). Compared to *Tch. tepidum* RC-LH1 that contains a closed LH1 ring, the B800s occupied the space of an N-terminal helix of LH1-α and the head of an ubiquinone (UQ) bound in the LHαβ interface (*Figure 2C*). Thus, incorporation of the B800s in nRC-LH occupied the LHαβ interface on the cytoplasmic side.

Notably, KγC in the LH ring of nRC-LH were located at two distinct positions (*Figures 1D and 2D*). 15 KγC$_{int}$ molecules obliquely spanned the LHαβ subunits, with the 4-oxo-β-ionone rings sandwiched between adjacent LHαβs and the $\phi$-end groups directed into the LH center. In addition, another 14 KγC were detected in a second position in the LH ring exterior (KγC$_{ext}$), which were almost parallel to the adjacent LHβ subunits; the 4-oxo-β-ionone rings were directed toward the cytoplasmic side

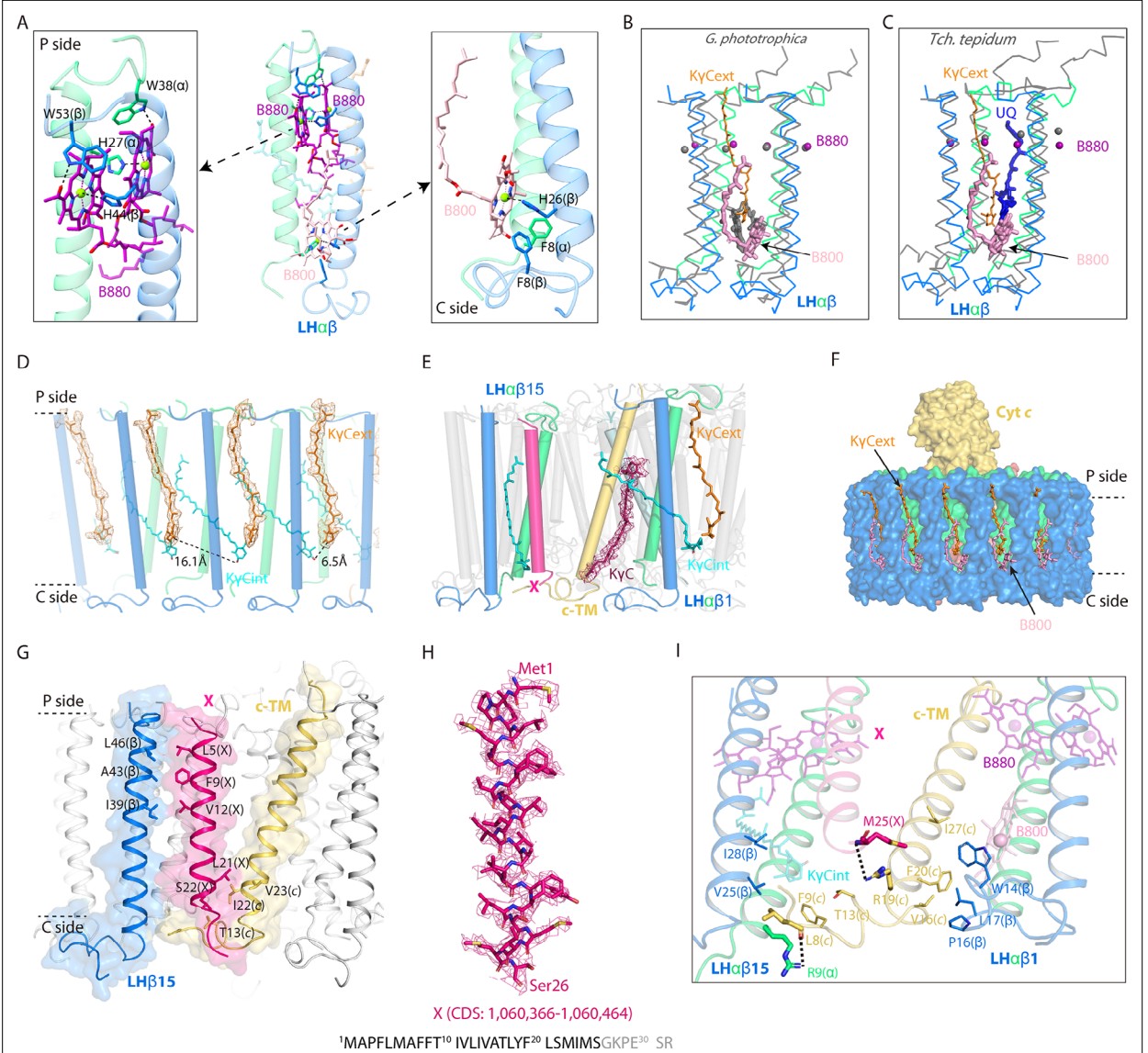

**Figure 2.** Interactions of the keto-γ-carotenes (KγC), bacteriochlorophylls (BChls), and subunit X with the light harvesting (LH) ring. (**A**) Interactions between the LHαβ heterodimer and the bound BChls. Close-up views of amino acid residues that coordinate the LH-bound B880s (left) and B800 (right) are shown on the periplasmic (P) and the cytoplasmic (C) side. The BChls and interacting amino acid residues are shown in stick forms. (**B, C**) Superposition of LHαβ heterodimer from nRC-LH (colored) with *Gemmatimonas* (*G.*) *phototrophica* LHh (B, gray) and *Tch. tepidum* LH1 (C, gray). The LH-bound B800 and exterior KγC (KγCext) in nRC-LH are shown as pink and orange sticks, respectively. Mg atoms of LH-bound B880 are shown in spheres. The LHh-bound B800 in *G. phototrophica* is shown in gray sticks, and *Tch. tepidum* LH1-bound ubiquinone (UQ) is shown in blue sticks. (**D, E**) KγC organization. Interior KγC (KγCint) are shown in cyan, KγCext are shown in orange, and the KγC inserted between cytochrome *c* transmembrane (c-TM) and LHαβ is shown in ruby. (**F**) Incorporation of the KγCext and B800s at the cytoplasmic side blocked the LHαβ interface. (**G, I**) Interactions between the assigned subunit X (hot pink), c-TM (yellow-orange), and neighboring LHαβ1 and LHαβ15 in the nRC-LH. The N-terminus (N-ter) and C-terminus (C-ter) of subunit X, c-TM and LHβ15 are indicated. The hydrogen bonding and hydrophobic interactions between the amino acid residues are labeled and indicated with dashed lines. The BChls B880 and B800 are shown as purple and pink sticks, respectively. (**H**) The assigned subunit X (hot pink) are fitted in the EM density map. Location of the coding sequence (CDS) in *R. castenholzii* genomic DNA, and the amino acid sequence of subunit X are indicated, with the modeled amino acid residues colored in black.

The online version of this article includes the following figure supplement(s) for figure 2:

**Figure supplement 1.** Interactions between LHαβ heterodimers in the native reaction center-light harvesting (RC-LH) (nRC-LH) complex from *R. castenholzii*.

**Figure supplement 2.** B-factors distribution of the native reaction center-light harvesting (nRC-LH) and carotenoid-depleted RC-LH (dRC-LH) complexes from *R. castenholzii*.

*Figure 2 continued on next page*

*Figure 2 continued*

**Figure supplement 3.** High-performance liquid chromatography (HPLC)-mass spectrometry (MS) analyses of the pigments in reaction center-light harvesting (RC-LH) complex from *R. castenholzii*.

**Figure supplement 4.** Structural comparison of the native reaction center-light harvesting (nRC-LH) from *R. castenholzii* with the RC-LH1s from *Rba. sphaeroides* and *Tch. tepidum*.

**Figure supplement 5.** Assignment of the amino acid sequences and coding sequences of subunit X and protein Y in *Roseiflexus* sp. and *R. castenholzii* DSM 13941/HLO8.

and the $\phi$-end groups stretched into the periplasm (*Figure 2D*). Alternatively, a newly identified KγC was sandwiched between LHαβ1 and c-TM, with its 4-oxo-β-ionone ring directing toward the RC-Y subunit (*Figure 2E*). The B-factor was higher for KγC$_{ext}$ than for KγC$_{int}$ molecules, with the latter having lower conformational flexibility (*Figure 2—figure supplement 2A*). Identification of these Cars yielded in a Car:BChl ratio of approximately 1:1.6 for the nRC-LH structure; this was consistent with results from previous pigment studies (*Collins et al., 2009*). High-performance liquid chromatography (HPLC)-mass spectrometry (MS) analyses of the pigments in nRC-LH revealed a typical BChl peak at the retention time of 5.58 min, and several peaks of γ-carotene and its derivatives (*Figure 2—figure supplement 3*). In respect to the complicated Car compositions and lack of specific absorption coefficients of the derivatives, it is impracticable to quantify the Car:BChl ratio from nRC-LH solution.

The nRC-LH thus resembled *Rba. sphaeroides* RC-LH1, which also binds two groups of Cars with different configurations (*Tani et al., 2021b*). Superposition analyses revealed similar Car positions and orientations between these two structures, although the keto groups of both Car types in nRC-LH were shifted toward the LHα subunits by ~6.7 Å (*Figure 2—figure supplement 4A and C*). Although KγC$_{ext}$ molecules were not well aligned with the LHαβ-bound UQ molecule in *Tch. tepidum* RC-LH1, they occupied the space between adjacent LHβs (*Figure 2C*, *Figure 2—figure supplement 4B and D*). As a result, the KγC$_{ext}$ molecules and additional B800s in *R. castenholzii* nRC-LH together blocked the LHαβ interface (*Figure 2F*), which serves as the quinone channel for the closed LH1 ring (*Qian et al., 2022*; *Yu et al., 2018b*), and for the opened LH1 ring bound only with interior Cars (*Qian et al., 2021a*; *Swainsbury et al., 2021*; *Yu et al., 2018b*).

## Assignment of the subunit X in nRC-LH complex

The *R. castenholzii* nRC-LH is distinguished from the RC-LH1 of most purple bateria by a newly identified subunit X and a membrane-bound cyt *c*, which has the TM helices that insert into the gap between LHαβ1 and LHαβ15 to form a putative quinone shuttling channel to the membrane quinone pool (*Xin et al., 2018*). Unlike the *Rba. sphaeroides* RC-LH1 protein PufX, which interacts with both LH1 and the L and H subunits of the RC (*Cao et al., 2022*; *Tani et al., 2022a*), subunit X in *R. castenholzii* was an independent TM helix that did not show any spatial overlap with PufX and PufY from the monomeric *Rba. sphaeroides* RC-LH1 (*Figure 2—figure supplement 4E*). Furthermore, compared with *Tch. tepidum* RC-LH1, which contains a closed LH1 ring, the c-TM of *R. castenholzii* nRC-LH was positioned close to the 16th LH1-α, whereas subunit X showed no overlap with the 16th LH1-β (*Figure 2—figure supplement 4B*). These structural features indicated that *R. castenholzii* RC-LH has evolved different structural elements to regulate quinone shuttling. However, the amino acid sequence of subunit X was unassigned in our previous 4.1 Å model, due to lack of clear cryo-EM densities.

From the high-resolution structure of nRC-LH, we successfully assigned the amino acid sequence (Met1-Ser26) for subunit X, which was derived from a hypothetical protein containing 32 amino acid residues (*Figure 2G and H*). This polypeptide was encoded by coding sequences (CDS: 1,060,366–1,060,464) in *R. castenholzii* (strain DSM 13941/HLO8) genome, but it was

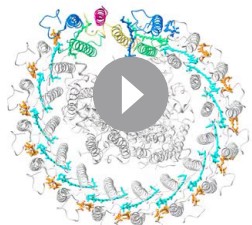

**Video 2.** Conformational changes of the light harvesting (LH) ring opening between native reaction center-LH (RC-LH) (nRC-LH) and carotenoid-depleted RC-LH (dRC-LH) complexes from *R. castenholzii*. The color scheme is same as *Figure 4C*.
https://elifesciences.org/articles/88951/figures#video2

not annotated in the Protein Database of Uniprot and NCBI. The amino acid sequence of subunit X showed strict conservation with a hypothetical protein KatS3mg058_1126 (GenBank: GIV99722.1) from *Roseiflexus* sp., which was denoted by metagenomic analyses of the uncultivated bacteria in Katase hot spring sediment (*Kato et al., 2022*; *Figure 2—figure supplement 5*). The resolved subunit X inserted into the LH opening in opposite orientation with that of LHαβ and c-TM, where these TM helices were stabilized by hydrophobic and weak hydrogen bonding interactions (*Figure 2G and I*). On the cytoplasmic side, the C-terminus of subunit X was coordinated in a pocket formed by the cyt *c* N-terminal region (Leu8, Phe9, and Thr13), LHβ15 (Val25 and Ile28), and the 4-oxo-β-ionone ring of a KγC$_{int}$ molecule. A weak hydrogen bond (3.5 Å) formed between the Met25 main chain nitrogen of subunit X and Arg19 amino nitrogen of c-TM. These pigment-protein interactions together stabilized the conformation of subunit X (*Figure 2I*, *Video 2*).

## Stabilizing the RC-LH interactions by newly assigned proteins Y and Z

Superposition of the RC structure with that of purple bacteria showed excellent matches at the L and M subunits, each of which contained five TM helices. Unlike purple bacteria, *R. castenholzii* L and M subunits are encoded by a fused gene *puf LM* but processed into two independent peptides in the complex (*Collins et al., 2010*; *Collins et al., 2009*; *Yamada et al., 2005*). In current model, RC-L subunit contains TM1-5 and terminates at Ala315, whereas the TM6-10 composed RC-M starts from Pro335 (*Figure 3A*, *Figure 3—figure supplement 1*, *Figure 3—figure supplement 2*). In addition, *R. castenholzii* RC-L contains an N-terminal extension (Met1-Pro35) that was solvent exposed on the cytoplasmic side (*Figure 3B*, *Figure 3—figure supplement 1A and C*). Most importantly, we resolved two additional TM helices in the RC (*Figure 3A*). Near the TM5 from RC-L and c-TM, a separate TM helix (corresponding to the TM7 in previous 4.1 Å model) was resolved with amino acid residues (Met1-Pro32) from a hypothetical protein Y (*Figure 3C*). Similar as subunit X, this protein was encoded by CDS (1,089,483–1,089,602) from *R. castenholzii* (strain DSM 13941/HLO8) genomic DNA, but it was not annotated in Protein Database as well. Coincidently, the amino acid sequence of protein Y was conserved with a hypothetical protein KatS3mg058_1154 (GenBank: GIV99750.1) from *Roseiflexus* sp. (*Figure 2—figure supplement 5*). The N-terminal region of protein Y was inclined toward the c-TM on the periplasmic side, wherein the 4-oxo-β-ionone ring of KγC was coordinated by hydrogen bonding interactions with Met11 (3.4 Å) from Y, Ser35 (3.0 Å) and Trp40 (2.8 Å) from the c-TM. On the cytoplasmic side, protein Y was stabilized by hydrogen bonding interactions with the TM5 of RC-L (*Figure 3B*).

Unlike purple bacteria, *R. castenholzii* RC does not contain an H subunit. Instead, we identified an individual TM helix between the LHα11 and RC-M (*Figures 1B and 3D–E*). Superposition revealed mismatch of this TM helix with that of the purple bacterial H subunit (*Figure 3A*). This helix was assigned to cover the amino acid residues Ser12 to Asn58 of a hypothetical protein (WP_041331144.1) from *R. castenholzii* (strain DSM 13941/HLO8) (*Figure 3D*), we named it protein Z. This protein was verified with a sequence coverage of 19% by peptide mass fingerprinting (PMF) analyses of the blue-native PAGE of the nRC-LH (*Table 1*). The resolved protein Z was stabilized by hydrogen bonding and hydrophobic interactions with amino acid residues from the RC-M and LHα11 on the periplasmic and cytoplasmic sides (*Figure 3E*).

In contrast with most purple bacteria, *R. castenholzii* cyt *c* contains an N-terminal transmembrane helix c-TM, which was absent in *G. phototrophica* and *Tch. tepidum* RC-bound cyt *c*, and was even distinct from *Rpi. globiformis* cyt *c* that also conains an N-terminal TM helix (*Tani et al., 2022b*; *Figure 3F*, *Figure 3—figure supplement 1B*, *Figure 3—figure supplement 3*). Compared to *Rpi. globiformis* cyt *c*, the c-TM was obliquely inserting into the LH opening in an opposite direction, wherein it formed a potential quinone shuttling channel with the subunit X (*Figure 3F*). The N-terminal cytoplasmic region of c-TM was stabilized by extensive hydrophobic interactions with LHαβ15 and LHαβ1 (*Figure 2I*). These included interactions between the cyt *c* Ile27, Phe20, and Val16 sidechains and the LHβ1 Trp14, Leu17, and Pro16 sidechains. The main chain oxygen of Leu8 formed a hydrogen bond with the guanidine nitrogen of Arg9 from LHα15 (3.2 Å). Notably, cyt *c* also formed extensive hydrogen bonding interactions with the RC-L and RC-M subunits at the heme3-binding region. In addition to the protein Y, Z, and cyt *c*-mediated interactions, another two close contact points were evident between the RC and LH: (i) helix 1 (TM1) from RC-L to LHα13, (ii) TM6 from RC-M to LHα4 and LHα5 (*Figure 3—figure supplement 4*). We also identified several structured lipids

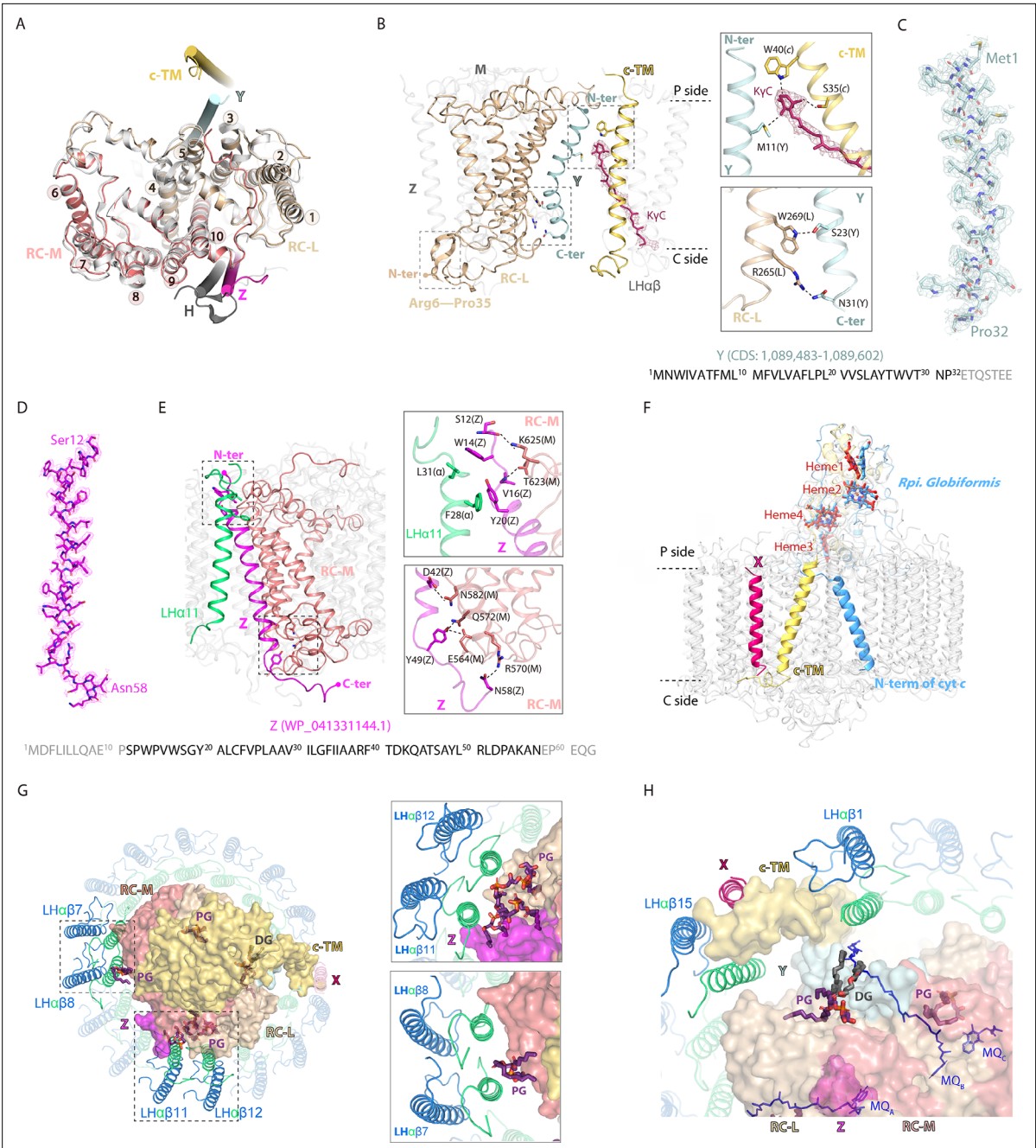

**Figure 3.** Stabilizing the reaction center (RC)-light harvesting (LH) interactions. (**A**) Superposition of *R. castenholzii* RC structure (colored) with that of *Rba. sphaeroides* (white, PDB ID: 7F0L) showed excellent match at the L (wheat) and M (salmon) subunits, each of which contains five transmembrane helices (TM1–5 for L and TM6–10 for M). The newly assigned TM helices from protein Y (pale cyan) and Z (light magenta) are located on the two sides of the RC. The only TM helix of *Rba. sphaeroides* H subunit (gray) does not match with that of protein Z. (**B**) Interactions between the assigned protein Y (pale cyan), cytochrome *c* transmembrane (c-TM) (yellow-orange), and the RC-L (wheat). The N-terminus (N-ter) and C-terminus (C-ter) of Y, and RC-L N-terminal extension (Arg6-Pro35) that located at the periplasmic (P) and the cytoplasmic (C) side are indicated. The hydrogen bonding interactions between the amino acid residues and KγC (ruby sticks) are labeled and indicated with dashed lines. (**C, D**) The assigned TM helix of protein Y (C, pale cyan) and protein Z (D, light magenta) are fitted in the electron microscopy (EM) density map. Location of the coding sequence (CDS) of Y in *R. castenholzii* genomic DNA and the protein accession number of protein Z are indicated. The amino acid sequences of protein Y and Z are indicated below, with the modeled amino acid residues colored in black. (**E**) Interactions between the assigned protein Z (light magenta), LHα11 (lime green), and the RC-M (salmon). The N-terminus (N-ter) and C-terminus (C-ter) of Z that located at the periplasmic (P) and the cytoplasmic (C) side are indicated. The hydrogen bonding interactions are shown in the dashed lines. (**F**) Superposition of *R. castenholzii* RC-bound cytochrome (cyt) *c* (yellow-orange) with that of *Rpi. globiformis* (cornflower blue, PDB ID: 7XXF) showed excellent match at the tetra-heme binding domain. The c-TM and N-ter of *Rpi. globiformis*

*Figure 3 continued on next page*

*Figure 3 continued*

cyt *c* directed into opposite directions. (**G, H**) Interactions of the lipids (phosphatidylglycerol, PG; and diglyceride, DG) with the LH and RC. The L, M, and cyt *c* subunits of RC are shown in surface, and LHαβs are shown in cartoon forms, the lipids and RC-bound menaquinone-11s (MQs) are shown in deep purple and blue sticks, respectively.

The online version of this article includes the following figure supplement(s) for figure 3:

**Figure supplement 1.** Comparison of *R. castenholzii* reaction center (RC) with the reported RCs from *Tch. tepidum* (PDB ID: 5Y5S), *G. phototrophica* (PDB ID: 7O0U), *Rpi. globiformis* (PDB ID: 7XXF).

**Figure supplement 2.** Structure-based sequence alignment of the M subunit from *R. castenholzii* and the representative purple bacteria.

**Figure supplement 3.** Structure-based sequence alignment of the cyt *c* subunit from *R. castenholzii* and the representative purple bacteria.

**Figure supplement 4.** Interactions between the light harvesting (LH) ring and reaction center (RC) in native RC-LH (nRC-LH) complex from *R. castenholzii*.

(phosphatidylglycerol, PG, and diglyceride, DG) within the interface between the RC and LH subunits (*Figure 3G and H*), these protein-lipids contacts further stabilized the nRC-LH complex.

## dRC-LH lacked subunit X

To explore the structural and functional relationships between LH-bound Cars and the RC-LH complex, *R. castenholzii* cells were photoheterotrophically cultured in the presence of DPA, a Car biosynthesis inhibitor (*Gall et al., 2005*). In response to DPA treatment, bacterial growth curves clearly indicated a decreased proliferation rate of cells grown under high illumination, confirming the important roles of Cars in photosynthesis and cell proliferation (*Figure 1—figure supplement 1A and B*). Interestingly, DPA treatment did not affect the growth of cells under medium and low illuminations, which showed an overall much lower proliferation rate (*Figure 1—figure supplement 1B*). Concomitantly, the color of the growing cells changed progressively from brownish red in the first culture to light yellow in the fifth sub-culture (*Figure 1—figure supplement 1A*), indicating gradual inhibition of Car biosynthesis during sub-culturing. To confirm the effects of DPA treatment on Car incorporation into the RC-LH, dRC-LH complexes were isolated from each successive sub-culture of DPA-treated *R. castenholzii* cells (*Figure 1—figure supplement 1F*). There was a striking decrease in Car absorbance in dRC-LH complexes extracted from the third through fifth sub-cultures of DPA-treated cells compared to nRC-LH extracted from untreated cells (*Figure 1—figure supplement 1D and E*). Additionally, HPLC analysis of dRC-LH isolated from the fifth sub-culture of DPA-treated cells showed same pigment compositions but strikingly decreased Car absorbance compared to the nRC-LH (*Figure 2—figure supplement 3B*).

To illustrate the effects of DPA treatment on the RC-LH architecture, we determined the cryo-EM structure of dRC-LH isolated from the fifth sub-culture of DPA-treated *R. castenholzii* cells at 3.1 Å resolution (*Figure 4A and B*, *Figure 4—figure supplement 1*). The most obvious difference between these two structures was the absence of the entire X subunit and the cytoplasmic region of cyt *c* subunit (Pro6-Val16) in the dRC-LH; both were located at the LH opening of nRC-LH (*Figure 4C*, *Video 2*, *Figure 1—figure supplement 6C*). Notably, only five KγC$_{int}$ molecules that spanned the LHαβ5, -7, -9, -10, and -11 heterodimers were resolved with clear density maps and built in the dRC-LH structure, whereas none of the KγC$_{ext}$ molecules were observed (*Figure 4B*, *Figure 1—figure supplement 4C*, *Figure 4—figure supplement 2A*, *Video 1*). The five KγC$_{int}$ molecules were located relatively far from the LH opening (~52 Å), which is where Cars with the highest B-factors were distributed, indicating an unstable conformation (*Figure 2—figure supplement 2A*). Additionally, the five KγC$_{int}$ molecules in dRC-LH adopted the same conformation and a similar edge-to-edge distance from LH-bound B800/B880s as the corresponding KγC$_{int}$ molecules did in nRC-LH (*Figure 4D*, *Tables 3 and 4*). The absence of KγC$_{ext}$ and most KγC$_{int}$ molecules in the LH ring confirmed the spectroscopic and HPLC analyses that DPA treatment decreased the numbers of LH-bound Cars in the dRC-LH.

To explore the effect of Car depletion on the LHαβ structure, we superposed the Car-bound LHαβ5, LHαβ7 with adjacent Car-unbound LHαβ6 and LHαβ8 in the dRC-LH. Except slight differences at the sidechain orientations of LHα-Phe28, these LHαβ heterodimers adopted exactly the same conformation (*Figure 4—figure supplement 2B*). However, both nRC-LH and dRC-LH contained the same LHα-Phe28 orientations at LHα7, -9, and -11. In addition, Phe28 sidechain orientations were not correlated with the Car binding, since each LHαβ in nRC-LH bound both KγC$_{int}$ and KγC$_{ext}$

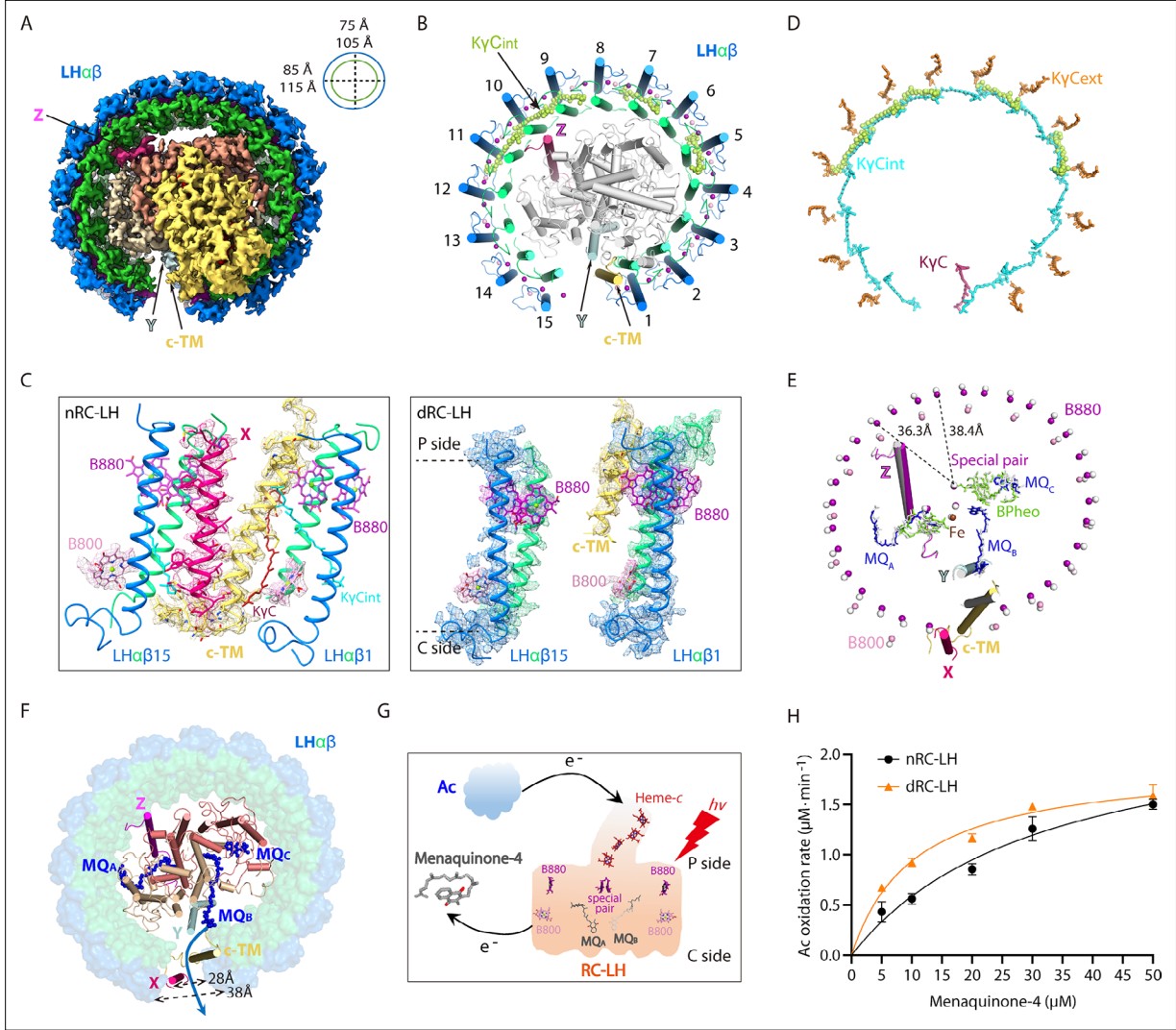

**Figure 4.** Cryo-electron microscopy (cryo-EM) structure of the carotenoid-depleted reaction center-light harvesting (dRC-LH) complex of *R. castenholzii* and its conformational changes that accelerated quinone/quinol exchange. (**A**) Cryo-EM map of dRC-LH seen from the bottom with LH ring dimensions indicated. (**B**) Cartoon representation of the dRC-LH complex from the bottom. The interior keto-γ-carotenes (KγCint) are shown in limon and bacteriochlorophyll (BChl) Mg atoms are shown as spheres. (**C**) Comparison of the LH ring openings in the nRC-LH (left) and dRC-LH (right). The cryo-EM maps of the subunit X (hot pink), cytochrome *c* transmembrane (c-TM) (yellow-orange), and neighboring LHαβ1 and LHαβ15 are shown to indicate the conformational changes. The LH-bound B880s (purple), B800s (pink), and KγC (ruby) are shown in sticks and fitted in the EM map. The periplasmic (P) and cytoplasmic (C) sides are labeled. (**D**) Comparison of the KγC arrangement between the nRC-LH and dRC-LH complexes. The KγCint, KγCext, and KγC in nRC-LH are shown as cyan, orange, and ruby sticks, respectively. The five KγCint molecules bound in the dRC-LH complex are shown as limon spheres. (**E**) Comparison of the central BChl-Mg atoms in nRC-LH and dRC-LH. The B880 and B800 Mg atoms are shown as purple and pink spheres, respectively, in nRC-LH, and as white spheres in dRC-LH. The two structures are superposed at the TM helices of the L and M subunits. The distances between the central Mg atoms of B880 and the nearest special pair BChls are labeled and indicated with dashed lines. The cofactors bound in the RC are shown in stick form; the iron is shown as spheres. TM helices of subunit X (hot pink), protein Y (pale cyan), Z (light magenta), c-TM (yellow-orange in nRC-LH and white in dRC-LH), and LHαβ1 and LHαβ15 (colored in nRC-LH and white in dRC-LH) are shown in ribbon form to demonstrate the spatial organization. (**F**) Comparison of the LH ring opening and quinone channels in nRC-LH and dRC-LH. The LH ring of dRC-LH is shown in surface form; the RC (including Y and Z), c-TM, and subunit X in nRC-LH are shown in cartoon forms; and menaquinones (MQs) are shown in blue sticks. Dashed lines indicate the dimensions of the LH ring openings in the two structures. The blue arrow represents the putative quinone shuttling path. (**G**) Model diagram of the auracyanin (Ac) oxidation assay. Upon illumination, light energy absorbed by the LH-bound BChls (B800 and B880) is transferred to RC. The primary charge separation occurs and initiates sequential electron transfer that reduces the MQs. The generated hydroquinone diffuses out of the RC-LH and exchanges with the menaquinone-4 in the solution. Once the reduced Ac is oxidized, the released electrons can be transferred back to reduce the photo-oxidized special pair through the *c*-type hemes. (**H**) The rate of Ac oxidation at various starting concentrations of menaquinone-4, in presence of the nRC-LH (black) or dRC-LH (orange). Data are shown as the mean ± standard deviations (n=3).

The online version of this article includes the following figure supplement(s) for figure 4:

*Figure 4 continued on next page*

**Figure supplement 1.** Cryo-electron microscopy (cryo-EM) analyses of the carotenoid-depleted reaction center-light harvesting (dRC-LH) complex from *R. castenholzii*.

**Figure supplement 2.** Structural comparisons of LHαβ heterodimers with bound and unbound keto-γ-carotenes (KγC) in *R. castenholzii* reaction center-light harvesting (RC-LH) complexes.

(*Figure 4—figure supplement 2C and D*). These observations thus indicated that Car depletion did not affect the LHαβ structure. Nevertheless, the distances between adjacent LHαs and LHβs in the dRC-LH showed average increases of 0.5 Å and 1.0 Å, respectively, compared with nRC-LH (*Table 5*). Accordingly, the Mg-to-Mg distances between adjacent B880s and B800s also increased in dRC-LH (*Tables 6 and 7*). Specifically, the LH-bound B880s and B800s shifted away from the LH ring center by ~2.0 Å, consequently increasing the Mg-to-Mg distance between LH-bound B880s and the nearest special pair of BChls in the RC (*Figure 4E*, *Table 8*). These results therefore indicated that Car depletion not only decreased the number of LH-bound Cars, but also altered the conformation of dRC-LH opening and pigments organizations. These alterations could affect the efficiency of energy transfer during the primary photochemical reactions (*Şener et al., 2011*; *Xin et al., 2012*).

## Conformational changes in the dRC-LH accelerated quinone/quinol exchange

In nRC-LH, insertion of the c-TM and subunit X at the LH opening, wherein the N-terminal cytoplasmic region of c-TM was stabilized by extensive hydrophobic and weak hydrogen bonding interactions with subunit X, LHαβ15, and LHαβ1 (*Figures 2I and 4C*). The c-TM was closer to LHα1 (9.7 Å) than to LHα15, whereas subunit X was closer to LHβ15 (11.2 Å), creating a narrow gap between the c-TM and the LHβ15 (*Figure 4C and F*, *Table 5*). The B800 pigment was not detected between c-TM and LHα15 (*Figure 4C*). Thus, the c-TM and subunit X were positioned to the sides of LHα1 and LHβ15, respectively; this formed a 19.4 Å gap between the c-TM and LHα15, and a 28 Å gap between subunit X and LHβ1, both of which may have allowed reduced quinones to exit the LH to the membrane quinone pool. Because dRC-LH lacked subunit X, the gap between LHβ1 and LHβ15 increased to ~38.0 Å (*Figure 4C and F*).

To investigate the functional effects of this conformational change, we compared the quinone/quinol exchange rates for nRC-LH and dRC-LH complexes. In the cyclic electron transport chain of *R. castenholzii*, the periplasmic electron acceptor auracyanin (Ac) transfers electrons back to the RC special pair through the tetra-heme of cyt *c* subunit, reducing the photo-oxidized special pair for turnover of the photo-reaction and electron transfer that subsequently reduce the bound menaquinones ($MQ_A$ and $MQ_B$) in the RC. The reduced $MQH_2$ is released from its binding site and exchanges with free MQs outside the RC-LH (*Figure 4G*). Using sodium dithionite-reduced Ac as the electron donor and menaquinone-4 as the electron acceptor, we measured Ac absorbance changes at 604 nm with varied concentrations of menaquinone-4 (*Figure 5—figure supplement 1A and B*). The initial oxidation rate of Ac was markedly higher in the presence of dRC-LH than nRC-LH (*Figure 4H*). This was consistent with the determined apparent Michaelis constants, which showed that dRC-LH had an accelerated quinone/quinol exchange rate of menaquinone-4 at 6.12±0.62 µM min⁻¹ (*Table 9*). The

**Table 3.** Edge-to-edge distance (Å) of interior keto-$\gamma$-carotenes (K$\gamma$C$_{int}$) to light harvesting (LH)-bound B800/B880s in the native reaction center-LH (nRC-LH) and carotenoid-depleted RC-LH (dRC-LH) complexes from *R. castenholzii*.

| nRC-LH | 1 | 2 | 3 | 4 | 5 | 6 | 7 | 8 | 9 | 10 | 11 | 12 | 13 | 14 | 15 |
|--------|---|---|---|---|---|---|---|---|---|----|----|----|----|----|----|
| B800 | – | 3.6 | 4.2 | 3.7 | 3.6 | 4.1 | 3.5 | 3.4 | 3.7 | 3.9 | 4.0 | 3.8 | 4.2 | 3.7 | – |
| B880 | – | 4.3 | 3.8 | 3.8 | 3.9 | 3.9 | 4.2 | 4.0 | 3.9 | 4.3 | 4.0 | 4.1 | 3.6 | 3.6 | 3.7 |
| dRC-LH | 1 | 2 | 3 | 4 | 5 | 6 | 7 | 8 | 9 | 10 | 11 | 12 | 13 | 14 | 15 |
| B800 | – | – | – | – | 4.2 | – | 3.4 | – | 4.1 | 3.3 | 3.9 | – | – | – | – |
| B880 | – | – | – | – | 4.0 | – | 4.2 | – | 4.4 | 3.8 | 4.0 | – | – | – | – |

**Table 4.** Edge-to-edge distance (Å) of exterior keto-$\gamma$-carotenes (K$\gamma$C$_{ext}$) to light harvesting (LH)-bound B800/B880s in the native reaction center-LH (nRC-LH) complex from *R. castenholzii.*

| nRC-LH | 1 | 2 | 3 | 4 | 5 | 6 | 7 | 8 | 9 | 10 | 11 | 12 | 13 | 14 | 15 |
|---|---|---|---|---|---|---|---|---|---|---|---|---|---|---|---|
| B800 | – | 3.7 | 3.4 | 3.9 | 3.8 | 3.8 | 3.7 | 3.3 | 3.5 | 3.2 | 3.3 | 3.4 | 3.5 | 3.5 | 4.2 |
| B880 | – | 4.8 | 5.0 | 5.0 | 5.0 | 4.1 | 4.2 | 4.3 | 4.8 | 4.8 | 5.1 | 4.7 | 4.2 | 4.8 | 4.6 |

accelerated quinone/quinol exchange rate in dRC-LH was probably resulted from exposure of the LHαβ interface by Car depletion, and also the increased gap dimension of the LH ring.

## Car depletion did not affect the Car-to-BChl energy transfer efficiency

To elucidate the effects of Cars depletion on the Car-to-BChl energy transfer efficiency of the RC-LH, we first examined the configurations and coordinating environments of the LH-bound Cars. K$\gamma$C$_{int}$ molecules spanned the TM region of each LHαβ heterodimer; the heads with 4-oxo-β-ionone ring were inserted into the hydrophobic pocket formed by the LHα and LHβ subunits, the phytol tails of two B880s, and the B800 porphyrin ring. On the periplasmic side, the $\psi$-end group of K$\gamma$C$_{int}$ was directed into a hydrophobic patch formed by two adjacent LHα subunits (*Figure 5A*, left). Alternatively, the newly identified K$\gamma$C$_{ext}$ molecules were immobilized in a position that was nearly parallel to the adjacent LHβs. The heads were inserted into a cavity formed by the B800 porphyrin ring and two adjacent LHβs, and their tails extended along the adjacent LHβs, stabilized by hydrophobic interactions (*Figure 5A*, right). However, depletion of these K$\gamma$C$_{ext}$ molecules in dRC-LH prevented the tight packing of the K$\gamma$C$_{int}$ molecules with LHαβ heterodimers. Thus, in the absence of K$\gamma$C$_{ext}$, the head of each K$\gamma$C$_{int}$ molecule shifted toward the B800 porphyrin ring, which moved the head out from the center of the LH ring by ~3.0 Å (*Figure 5B*). However, the edge-to-edge distances of K$\gamma$C$_{int}$ to the B800/B880s remained similar between dRC-LH and nRC-LH (*Table 3*).

We next measured the fluorescence excitation and absorption spectra of the nRC-LH and dRC-LH complexes to calculate the Car-to-BChl energy transfer efficiency. Most RC-LH fluorescence is emitted from the B880 Qy band (*Collins et al., 2009*). Excitation of nRC-LH at 470 nm yielded emissions at 900 nm, whereas dRC-LH excitation produced emissions at 905 nm (*Figure 5C*). This shift of the emission peak indicated changes in the LH ring pigment configuration between the two complexes. The intensity ratio of fluorescence excitation spectra to absorption spectra, expressed as the 1−T of RC-LH, was then calculated. The results revealed that the Car-to-BChl energy transfer efficiency remained similar between nRC-LH (44%) and dRC-LH (46%) (*Figure 5D*). Car-to-BChl energy transfer in the LH is closely related to the number of Car conjugated double bonds, the relative distances between Cars and BChls, and Car/BChl spatial organization (*Polívka and Frank, 2010*). In *R. castenholzii*, each K$\gamma$C contains 11 conjugated double bonds (*Collins et al., 2009*). Although all K$\gamma$C$_{ext}$ and most K$\gamma$C$_{int}$ molecules were depleted in dRC-LH, the five remaining K$\gamma$C$_{int}$ molecules adopted the same configuration and similar edge-to-edge distances with LH-bound B800/B880s as that in the nRC-LH (*Figure 4D*, *Table 3*). Therefore, Car depletion from the LH ring in dRC-LH did not affect interactions between the remaining Cars and BChls, which exhibited similar excitation energy transfer values in dRC-LH and

**Table 5.** Distances (Å) between the LHαβ transmembrane (TM) helices of the native reaction center-light harvesting (nRC-LH) and carotenoid-depleted RC-LH (dRC-LH) complexes from *R. castenholzii.* The distance is measured between LHαβn and LHαβ (n+1).

| nRC-LH | 1 | 2 | 3 | 4 | 5 | 6 | 7 | 8 | 9 | 10 | 11 | 12 | 13 | 14 | 15* | 16† |
|---|---|---|---|---|---|---|---|---|---|---|---|---|---|---|---|---|
| LHα | 14.8 | 14.7 | 14.6 | 14.3 | 14.4 | 14.7 | 15.4 | 14.9 | 15.4 | 14.6 | 14.7 | 15.0 | 14.5 | 14.4 | 19.4 | 9.7 |
| LHβ | 20.0 | 19.8 | 20.0 | 20.4 | 20.2 | 20.0 | 19.8 | 19.9 | 19.7 | 20.1 | 20.0 | 19.9 | 20.1 | 20.1 | 11.2 | 28.1 |
| dRC-LH | 1 | 2 | 3 | 4 | 5 | 6 | 7 | 8 | 9 | 10 | 11 | 12 | 13 | 14 | 15* | 16† |
| LHα | 15.4 | 15.4 | 15.3 | 15.0 | 14.9 | 15.3 | 16.2 | 15.5 | 16.1 | 15.2 | 15.4 | 15.6 | 15.3 | 14.9 | 19.6 | 10.2 |
| LHβ | 21.0 | 20.7 | 20.9 | 21.2 | 20.9 | 21.0 | 20.5 | 20.9 | 20.6 | 20.8 | 21.0 | 20.7 | 20.9 | 20.9 | – | – |

*The distances from LHα15 Met-22 to c-TM Val-33, and LHβ15 Ile-39 to subunit X.
†The distances from LHα1 Val-29 to c-TM Tyr-44, and LHβ1 Leu-37 to subunit X.

**Table 6.** The distances (Å) between adjacent light harvesting (LH)-bound B880s of the native reaction center-LH (nRC-LH) and carotenoid-depleted RC-LH (dRC-LH) complexes from *R. castenholzii*.

| nRC-LH | 1 | 2 | 3 | 4 | 5 | 6 | 7 | 8 | 9 | 10 | 11 | 12 | 13 | 14 | 15 |
|---|---|---|---|---|---|---|---|---|---|---|---|---|---|---|---|
| Edge-to-edge* | 3.3 | 3.1 | 3.4 | 3.4 | 3.6 | 3.8 | 3.7 | 3.8 | 3.7 | 3.6 | 3.5 | 3.6 | 3.5 | 3.6 | 3.8 |
| Edge-to-edge† | 3.5 | 3.4 | 3.6 | 3.6 | 3.4 | 4.0 | 3.6 | 3.8 | 3.5 | 3.5 | 3.3 | 3.7 | 3.6 | 3.6 | 19.1 |
| Mg-to-Mg* | 10.2 | 9.9 | 9.8 | 9.9 | 9.7 | 9.8 | 9.9 | 10.0 | 9.9 | 9.9 | 9.8 | 9.6 | 9.7 | 9.3 | 9.8 |
| Mg-to-Mg† | 7.6 | 7.5 | 7.7 | 7.7 | 7.6 | 7.6 | 7.6 | 7.5 | 7.6 | 7.8 | 7.7 | 7.9 | 7.9 | 7.9 | 26.2 |
| **dRC-LH** | **1** | **2** | **3** | **4** | **5** | **6** | **7** | **8** | **9** | **10** | **11** | **12** | **13** | **14** | **15** |
| Edge-to-edge* | 3.5 | 3.5 | 3.5 | 3.7 | 3.5 | 3.7 | 3.7 | 3.8 | 3.9 | 3.8 | 3.6 | 3.5 | 3.6 | 3.3 | 3.7 |
| Edge-to-edge† | 4.1 | 3.8 | 4.0 | 3.8 | 3.9 | 3.8 | 4.1 | 4.0 | 4.1 | 3.9 | 3.9 | 3.9 | 3.9 | 3.8 | 20.2 |
| Mg-to-Mg* | 9.7 | 9.7 | 9.5 | 9.6 | 9.6 | 9.6 | 9.4 | 9.7 | 9.7 | 9.6 | 9.5 | 9.4 | 9.5 | 9.3 | 9.2 |
| Mg-to-Mg† | 9.0 | 8.7 | 8.8 | 8.8 | 8.4 | 8.9 | 8.8 | 8.6 | 8.8 | 8.8 | 8.9 | 9.0 | 9.0 | 8.7 | 27.2 |

*Distance between the B880 bound by the same transmembrane pairs of LH.
†Distance between the B880 bound by adjacent transmembrane pairs of LH.

nRC-LH complexes. These results suggested that the existing Car-to-BChl energy transfer efficiency is similar even though there is variation in the number of LH-bound Cars.

## Discussion

Unlike the well-studied purple bacteria, which contain two types of LH complexes, *R. castenholzii* contains only one RC-LH complex for LH and primary photochemical reactions. It does not contain the H subunit that is typically found in purple bacteria (*Pugh et al., 1998*; *Qian et al., 2005*; *Yamada et al., 2005*). Especially, *R. castenholzii* RC-LH contains a tetra-heme cyt *c* subunit that interrupts the LH ring, which is composed of 15 αβ-polypeptides, through a novel N-terminal TM helix; together with the newly identified subunit X, this forms a potential quinone shuttling channel on the LH ring. In the present study, we determined high-resolution cryo-EM structures of nRC-LH, from which we assigned the full amino acid sequence of subunit X, and two additional TM helices derived from hypothetical proteins Y and Z in the RC, which both functioned in stabilizing the RC-LH interactions. Most importantly, we identified 14 additional KγC molecules (KγC$_{ext}$) in the LH ring exterior, and one KγC inserted between LHαβ1 and c-TM, which generated a 2:3 Car:BChl molar ratio consistent with previous pigments analyses (*Collins et al., 2009*). Binding of the KγC$_{int}$ and KγC$_{ext}$ together with the B800s blocked the proposed quinone channel between LHαβ subunits. DPA treatment of the cells yielded a dRC-LH, referred to as dRC-LH; a 3.1 Å resolution cryo-EM structure resolved only five KγC$_{int}$ molecules, and the absence of subunit X and the cytoplasmic region of c-TM. These alterations in the dRC-LH increased the size of the LH opening and exposed the LHαβ interface, accelerating the in vitro quinone/quinol exchange rate of menaquinone-4, but did not affect the Car-to-BChl energy transfer efficiency.

**Table 7.** The distances (Å) between light harvesting (LH)-bound B800s of the native reaction center-LH (nRC-LH) and carotenoid-depleted RC-LH (dRC-LH) complexes from *R. castenholzii*.

| nRC-LH | 1 | 2 | 3 | 4 | 5 | 6 | 7 | 8 | 9 | 10 | 11 | 12 | 13 | 14 | 15 |
|---|---|---|---|---|---|---|---|---|---|---|---|---|---|---|---|
| Edge-to-edge | 15.4 | 15.5 | 15.5 | 15.4 | 15.4 | 15.6 | 15.6 | 15.4 | 15.6 | 15.5 | 15.5 | 15.6 | 15.6 | 15.2 | 33.4 |
| Mg-to-Mg | 18.4 | 18.3 | 18.4 | 18.5 | 18.5 | 18.5 | 18.5 | 18.3 | 18.4 | 18.4 | 18.5 | 18.5 | 18.4 | 18.3 | 38.5 |
| **dRC-LH** | **1** | **2** | **3** | **4** | **5** | **6** | **7** | **8** | **9** | **10** | **11** | **12** | **13** | **14** | **15** |
| Edge-to-edge | 15.8 | 15.9 | 16.0 | 15.7 | 15.8 | 16.2 | 15.9 | 15.8 | 16.1 | 15.8 | 16.0 | 16.1 | 15.8 | 15.8 | 33.2 |
| Mg-to-Mg | 19.1 | 19.1 | 19.3 | 19.1 | 19.2 | 19.4 | 19.1 | 19.2 | 19.2 | 19.1 | 19.3 | 19.3 | 19.1 | 19.2 | 38.3 |

**Table 8.** The Mg-to-Mg distances between light harvesting (LH)-bound B880s and the nearest special pair of bacteriochlorophyll (BChls) in the reaction center (RC) of *R. castenholzii* native RC-LH (nRC-LH) and carotenoid-depleted RC-LH (dRC-LH) complexes.

| nRC-LH | 1 | 2 | 3 | 4 | 5 | 6 | 7 | 8 | 9 | 10 | 11 | 12 | 13 | 14 | 15 |
|---|---|---|---|---|---|---|---|---|---|---|---|---|---|---|---|
| Mg-to-Mg* | 40.3 | 42.4 | 44.7 | 47.2 | 46.8 | 43.9 | 40.2 | 37.3 | 36.5 | 37.7 | 40.6 | 43.7 | 44.5 | 44.8 | 44.2 |
| Mg-to-Mg† | 41.5 | 43.8 | 46.3 | 48.1 | 45.6 | 42.0 | 38.5 | 36.8 | 36.8 | 39.6 | 42.6 | 44.5 | 44.9 | 44.8 | 43.6 |
| **dRC-LH** | 1 | 2 | 3 | 4 | 5 | 6 | 7 | 8 | 9 | 10 | 11 | 12 | 13 | 14 | 15 |
| Mg-to-Mg* | 42.0 | 44.8 | 47.2 | 49.9 | 49.0 | 45.8 | 42.0 | 39.1 | 38.3 | 39.7 | 43.0 | 46.0 | 46.2 | 46.4 | 45.4 |
| Mg-to-Mg† | 43.1 | 45.8 | 48.5 | 50.0 | 47.2 | 43.8 | 40.2 | 38.4 | 38.6 | 41.3 | 44.5 | 46.0 | 46.3 | 46.1 | 44.6 |

*The distances from $Mg^{2+}$ of the first LH-bound B880 to the nearest special pair of BChls.
†The distances from $Mg^{2+}$ of the second LH-bound B880 to the nearest special pair of BChls.

To maintain continuous photo-reaction and turnover of the electron transport chain, two quinone exchange/transport routes are required for the bacterial RC-LH1 complex. One is the exchange route for the free/bound quinone in the RC, which was represented by a newly identified MQc in our nRC-LH structure (*Figure 3H*), and also extra UQ molecules found in many purple bacterial RCs (*Cao et al., 2022*; *Kishi et al., 2021*; *Qian et al., 2022*; *Qian et al., 2018*; *Swainsbury et al., 2021*; *Tani et al., 2022b*; *Yu et al., 2018a*). The other one is shuttling channel between the inside and outside of the LH1 ring. For *Tch. tepidum* RC-LH1 that contains an almost symmetric and completely closed LH1 ring, except the 'waiting' UQ8 identified near $Q_B$, one UQ8 was found to be inserted between the LH1α and LH1β subunits (*Yu et al., 2018a*), representing a potential quinone exchange channel between the LHαβ interface. Therefore, the space between the LHαβ subunits can serve as quinone exchange channel for the closed LH1 ring (*Qian et al., 2022*; *Yu et al., 2018b*), and also for the opened LH1 ring bound only with interior Cars (*Qian et al., 2021a*; *Swainsbury et al., 2021*; *Yu et al., 2018b*). For most purple bacterial RC-LH1 complexes with an opened C-shaped LH1 ring, reduced quinones are also shuttled from the RC through a gap at the LH1 ring, which is disrupted by protein W, or PufX and PufY (or protein-U) (*Cao et al., 2022*; *Jackson et al., 2018*; *Qian et al., 2021a*; *Tani et al., 2021a*; *Tani et al., 2022a*; *Tani et al., 2021b*).

Distinct from the RC-LH1 of most purple bacteria, each LHαβ of *R. castenholzii* non-covalently bound an additional B800 BChl at the cytoplasmic side, which occupied the LHαβ interface at the cytoplasmic side (*Figure 2C*, *Figure 2—figure supplement 1C*). In addition, we identified KγC at three distinct positions in the nRC-LH ring: $KγC_{int}$ and $KγC_{ext}$, and also an additional KγC near the LH opening (*Figures 1D and 3B*). The $KγC_{int}$ molecules embedded between the LHαβs had a similar conformation as they do in the completely closed and also the opened LH1 ring of purple bacteria. In contrast, $KγC_{ext}$ molecules occupied the space between adjacent LHβs, although they were not well aligned with the *Tch. tepidum* LHαβ-bound UQ8 molecule (*Figure 2C*, *Figure 2—figure supplement 4D*). Therefore, incorporation of the $KγC_{ext}$ molecules and additional B800s in *R. castenholzii* nRC-LH most likely together blocked the LHαβ interface for putative quinone exchange (*Figure 2F*). Alternatively, *R. castenholzii* RC-LH incorporated a membrane-bound cyt *c* and a hypothetical protein X, which has the TM helices that interrupted the LH ring to form a potential channel for controlled quinone/quinol exchange (*Figure 6*). Superposition of *R. castenholzii* with purple bacterial RC-LH1s revealed distinct locations and orientations of subunit X and c-TM compared to PufX and PufY

**Table 9.** Apparent Michaelis constants of menaquinone-4 as electron acceptor in the auracyanin (Ac) oxidation assay, in presence of the *R. castenholzii* native reaction center-light harvesting (nRC-LH) or carotenoid-depleted RC-LH (dRC-LH) complex.

|  | nRC-LH | dRC-LH |
|---|---|---|
| $K_m$ (µM) | 32.24±6.84 | 10.43±1.23 |
| $k_{cat}$ ($min^{-1}$) | 82.61±9.09 | 63.83±2.49 |
| $k_{cat}/K_m$ | 2.56±0.49 | 6.12±0.62 |

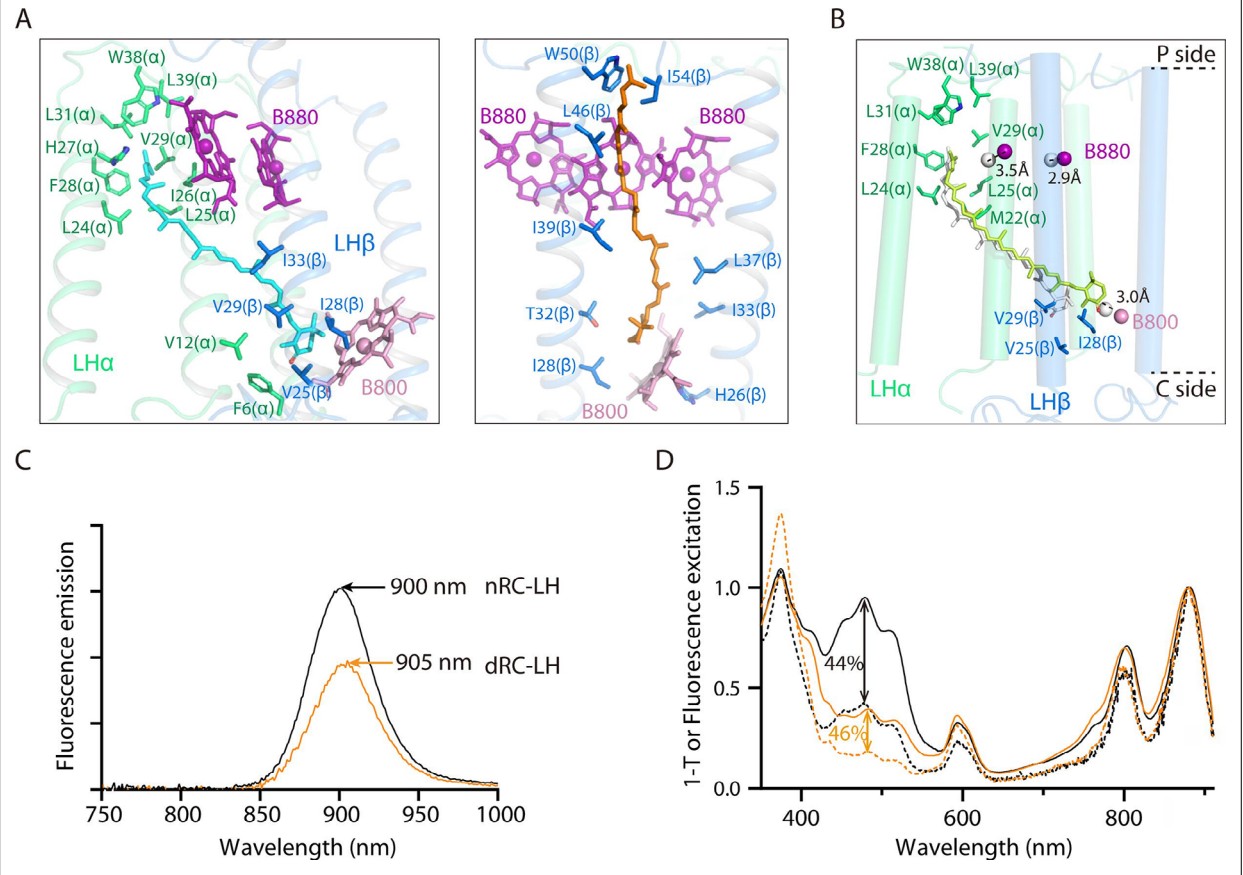

**Figure 5.** Binding conformation of the interior and exterior keto-γ-carotenes (KγC$_{int}$ and KγC$_{ext}$, respectively) and measurement of the Car-to-BChl energy transfer efficiency in native reaction center-light harvesting (nRC-LH) and carotenoid-depleted RC-LH (dRC-LH) complexes. (**A**) Coordination of representative KγC$_{int}$ (cyan) and KγC$_{ext}$ (orange) molecules in the nRC-LH complex. Shown as stick forms are the amino acid residues from LHα (lime green) and LHβ (marine) surrounding the 4-oxo-β-ionone ring; the $\phi$-end group of the KγC; and the BChls B880 (purple) and B800 (pink) in the nearby LHαβ. (**B**) Coordination of the KγC$_{int}$ molecules, which are shown in limon and white in dRC-LH and nRC-LH, respectively. Amino acid residues from the nearby LHα (lime green) and LHβ (marine) and the B800 molecule that covers the KγC$_{int}$ molecule are shown as stick forms. The distance deviations of the central Mg atoms in B880 (purple) and B800 (pink) in the two structures are labeled and indicated with dashed lines. The periplasmic (P) and cytoplasmic (C) sides are labeled. (**C**) Spectral analysis of the RC-LH complex. Fluorescence emissions are shown for nRC-LH (black) and dRC-LH (orange) complexes isolated from *R. castenholzii* after excitation at 470 nm. (**D**) Fluorescence excitation and absorption (1−T) spectra are shown as dotted and solid lines, respectively, for nRC-LH (black) and dRC-LH (orange). The Car-to-BChl energy transfer efficiency (vertical dashed line) was calculated by normalizing the fluorescence excitation and absorption spectra at 880 nm to 1.0.

The online version of this article includes the following figure supplement(s) for figure 5:

**Figure supplement 1.** The auracyanin (Ac) oxidation activities of the native reaction center-light harvesting (nRC-LH) and carotenoid-depleted RC-LH (dRC-LH) complexes from *R. castenholzii*.

(*Figure 2—figure supplement 4E*), indicating *R. castenholzii* has evolved different structural elements for regulating quinone shuttling.

Genetic depletion of the LH1-bound Cars promoted the photosynthetic growth of a PufX-knockout *Rba. sphaeroides* mutant with a closed LH1 ring (*Cao et al., 2022*; *McGlynn et al., 1994*; *Olsen et al., 2017*); this implies that disruption of Cars binding exposed the blocked quinone channel between LHαβ interface and facilitated the quinone exchange, thus promoting photosynthetic growth. In our study, depletion of the KγC$_{ext}$ and most KγC$_{int}$ molecules by DPA treatment could also expose the space between the Car-unbound LHαβ subunits. In addition, absence of the subunit X and cytoplasmic region of c-TM in dRC-LH broadened the dimensions of the LH ring opening, which most likely together accelerated the quinone/quinol exchange rate of the dRC-LH (*Figure 6*). This was consistent with a previous observation that the open form of the *Rhodopseudomonas* (*Rps.*) *palustris* RC-LH1 has a faster UQ2 diffusion rate than the closed form (*Swainsbury et al., 2021*). Notably,

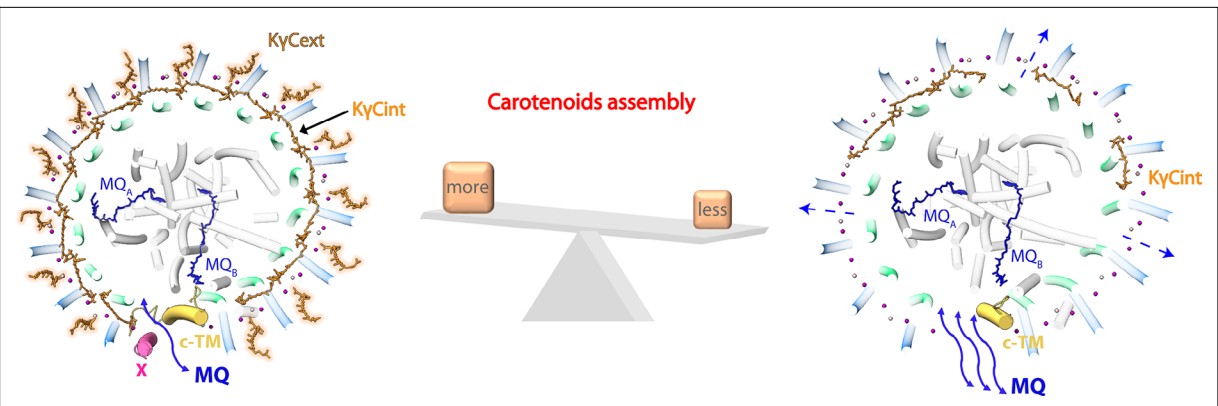

**Figure 6.** Schematic diagram of the carotenoid (Car) assembly-related structural dynamics of *R. castenholzii* reaction center-light harvesting (RC-LH) complex. In native RC-LH, incorporation of the external keto-γ-carotenes (KγC$_{ext}$) and LH-bound B800s blocked the LHαβ interface. Alternatively, the subunit X disrupts the ring and forms a potential quinone channel with the cytochrome *c* transmembrane (c-TM), facilitating controlled quinone/quinol binding and shuttling. In Car-depleted RC-LH (dRC-LH), less Car assembly exposed the LHαβ interface, absence of the subunit X and cytoplasmic region of c-TM concomitantly broadened the LH opening, which together accelerated the quinone/quinol exchange.

depletion of most LH-bound Cars only affected the stable conformation of the cytoplasmic region of c-TM, which was closely associated with subunit X to form the putative quinone channel (*Figure 2I*). Compared to cyt *c* subunit that formed extensive hydrogen bonding interactions with the L, M, and Y of the RC, the subunit X was characterized by high B-factors, fewer contacts with the RC-LH, and an easily disrupted conformation (*Figures 3B and 4C*, *Figure 2—figure supplement 2B*). Especially, the subunit X was derived from a hypothetical protein that inserted into the LH opening in an opposite orientation with LHαβ and c-TM, suggesting that it was likely the last subunit incorporated into the RC-LH. Therefore, *R. castenholzii* RC-LH could probably evolve the subunit X to control the conformation of the quinone shuttling channel.

Cars contribute to the self-assemble of natural α/β polypeptides to form LH1 complexes in vitro (*Fiedor et al., 2004*), Car-less *Rsp. rubrum* LH1 can be obtained by exogenous recombination (*Parkes-Loach et al., 1988*). In the Car-less *Rba. sphaeroides* mutant strain R26, the polymerized form of RC-LH is predominantly monomeric, and the curvature of the photosynthetic membrane is altered due to the lack of dimeric RC-LH (*Ng et al., 2011*). This implies that Cars assembly can regulate the conformation of the RC-LH complex. In our study, Car depletion also affected the LH opening conformation and the quinone/quinol exchange rate of the dRC-LH. Although the extensive interactions between subunit X, c-TM, and LHαβ15 and LHαβ1 were disrupted in dRC-LH (*Figures 2I and 4C*), the correlation between Car depletion and the absence of subunit X has not been adequately verified. Since DPA treatment is not a clean way to examine the effect of Cars, it left several interior Cars still bound to the LH ring. DPA is a broad-spectrum inhibitor that slows cellular metabolic processes and specifically affects Car biosynthesis by inhibiting phytoene desaturase (CrtI), an essential enzyme catalyzes conversion of the colorless Car precursor phytoene to the colored lycopene (*Bramley, 1993*). We here found that DPA treatment not only dramatically decreased the *R. castenholzii* proliferation rate but also depleted the LH-bound Cars in dRC-LH (*Figures 1A and 4*, *Figure 1—figure supplement 1B*). However, an efficient genetic manipulation system of *R. castenholzii* is required to obtain a Car-less RC-LH complex, for elucidating the correlations between Cars and the RC-LH assembly, as well as the photosynthetic growth of cells. To our current knowledge, genetic editing of *R. castenholzii* is restricted by its morphology as a multicellular filamentous bacterium with an optimal growth temperature ~50°C, and the lack of a well-studied genetic background that facilitates exogenous DNA introduction and replication.

In summary, this study revealed conformational changes of the *R. castenholzii* RC-LH in the presence and absence of KγC$_{ext}$ and subunit X, which played a role in regulating the quinone/quinol exchange. KγC$_{ext}$ incorporation results in a sealed conformation of the LH ring, whereas Car depletion and absence of the subunit X produces an exposed LH ring with larger opening, which together accelerate the in vitro quinone/quinol exchange of menaquinone-4. These results indicate a correlation between LH-bound Cars and the assembly and quinone/quinol exchange of *R. castenholzii* RC-LH.

Overall, these findings deepen our understanding of the light absorption and photo-reaction mechanisms in prokaryotic photosynthesis and increase the feasibility of applying prokaryotic photosystems in synthetic microbiology approaches.

## Materials and methods

### Key resources table

| Reagent type (species) or resource | Designation | Source or reference | Identifiers | Additional information |
|---|---|---|---|---|
| Strain, strain background (*Roseiflexus castenholzii*) | DSM 13941/HLO8 | *Hanada et al., 2002* | / | / |
| Chemical compound, drug | *N*-Dodecyl-β-D-maltoside (β-DDM) | Anatrace | D310 | / |
| Chemical compound, drug | Bacteriochlorophyll *a* | Sigma-Aldrich | B5906 | / |
| Chemical compound, drug | γ-Carotene | Sigma-Aldrich | 54765 | / |
| Chemical compound, drug | Menaquinone-4 | Sigma-Aldrich | PHR2271 | / |
| Software, algorithm | RELION 3.1 | *Zivanov et al., 2018* | / | / |

### Extraction and purification of the RC-LH complexes from *R. castenholzii*

The *R. castenholzii* cells (strain DSM 13941/HLO8) were grown anaerobically at 50°C under high (180 µmol m$^{-2}$ s$^{-1}$), medium (32 µmol m$^{-2}$ s$^{-1}$), and low (2 µmol m$^{-2}$ s$^{-1}$) illuminations in a modified PE medium as previously reported (*Hanada et al., 2002*). To inhibit Car biosynthesis, DPA was added to the medium (12 mg L$^{-1}$), and the bacteria were cultured under the same conditions as the native bacteria. Growth curves of the native and DPA-treated *R. castenholzii* cells were monitored with a UV-vis spectrophotometer (Mapada P6, Shanghai), by recording the absorption of cultured cells at 660 nm for every 12 hr. The mean values of the optical density at each time point and the standard deviations of mean (n=3) were calculated.

Isolation and purification of both the nRC-LH and dRC-LH complexes were carried out as described (*Collins et al., 2009*) with some modifications. The whole membranes (OD = 20 cm$^{-1}$ measured at 880 nm) in 20 mM Tris-HCl (pH 8.0) were solubilized by 0.45% (wt/vol) β-DDM (Anatrace, USA) at room temperature for 1 hr with gentle stirring and then were ultracentrifuged at 200,000×*g* for 1 hr. The supernatant was collected and filtered through a 0.22 µm filter and diluted with buffer A (0.04% β-DDM, 50 mM Tris-HCl, pH 8.0), subsequently loaded on an anion exchange chromatography column (HiTrap Q HP, Cytiva, USA) that had been equilibrated with buffer A. The crude RC-LH complex was eluted from the column with 200 mM NaCl in buffer A, and further purified by gel filtration through a Superdex 200 16/600 column, and a Superose 6 Increase 10/300 GL (Cytiva, USA) in buffer B (0.04% β-DDM, 100 mM NaCl, 50 mM Tris-HCl, pH 8.0). The whole preparation procedure was monitored by detecting the absorption spectrum from 250 to 900 nm.

### HPLC-MS analyses of the pigments in RC-LH complexes

Pigment composition was analyzed by HPLC as described (*Collins et al., 2009*). The RC-LH samples were mixed with acetone/methanol (vol/vol ratio of 7:2) to extract the pigments, followed by centrifugation at 12,000×*g* for 15 min. Then, the supernatant was filtered through a 0.22 µm filter membrane. The filtrate was injected into a C18 reversed-phase column (4.6 mm×150 mm, 5 µm particle size, Agilent, USA) in a Thermo Fisher Ultimate 3000 separation module equipped with a DAD-3000 Diode Array Detector. The pigments were eluted at a flow rate of 1 mL min$^{-1}$ using 100% methanol. Pigments were then detected by their absorbance at 462 nm and 772 nm. The commercial BChl *a* and γ-carotene (Sigma-Aldrich, USA) were used as standards. Pigments were identified based on their absorption spectra, retention times, and further analyzed by LC-MS. LC-MS was equipped with an Agilent 1200 HPLC system (Agilent, Santa Clara, CA. USA) and a Thermo Finnigan LCQDeca XP Max LC/MS system (Thermo Finnigan, Waltham, MA, USA). The condition of HPLC is the same as the above. MS with an atmospheric pressure chemical ionization source was performed as follows: positive mode, source voltage of 2.5 kV, capillary voltage of 46 V, sheath gas flow of 60 arbitrary units, auxiliary/sweep gas flow of 10 arbitrary units, capillary temperature 150°C. The pigments composition was determined as shown in *Figure 2—figure supplement 3*.

## Cryo-EM

Three µL aliquots of the purified RC-LH (native and Car-depleted) complexes were placed on glow-discharged CryoMatrix R1.2/1.3 300-mesh amorphous alloy film (product no. M024-Au300-R12/13, Zhenjiang Lehua Technology Co. Ltd., China). Each grid was blotted for 3 s at 4°C in 100% humidity, then plunged into liquid ethane with a Mark IV Vitrobot system (Thermo Fisher Scientific, USA).

Data for the nRC-LH complex was collected on a 300 kV Titan Krios electron microscope (Thermo Fisher Scientific, USA) with a K3 direct electron detector (Gatan, USA) in counting mode. A total of 2,836 movies were recorded at a magnification of ×64,000 and a pixel size of 1.08 Å, with a total dose of approximately 50 e$^-$ Å$^{-2}$, and a defocus range between –1.0 and –2.3 µm. Each movie was collected over 2.59 s and dose-fractionated into 40 frames. Data for the dRC-LH complex was recorded on a 300 kV Titan Krios electron microscope with a K3 direct electron detector in counting mode. A nominal magnification of ×81,000 was used for imaging, which yielded a pixel size of 0.893 Å. A total of 3,514 movies were collected with defocus values between –1.1 µm and –1.7 µm. Each movie was dose-fractionated to 40 frames under a total dose of 49.65 e$^-$ Å$^{-2}$ and an exposure time of 2.2 s. Cryo-EM analyses of nRC-LH complexes extracted from cells grown under medium (32 µmol m$^{-2}$ s$^{-1}$) and low (2 µmol m$^{-2}$ s$^{-1}$) illuminations were summarized in *Figure 1—figure supplement 3* and *Table 2*.

## Image processing

Beam-induced motion correction and exposure weighting were performed by MotionCorr2 (*Zheng et al., 2017*), and the CTF (contrast transfer function) was estimated using the Gctf program (*Zhang, 2016*). The automatic particle picking was performed by Gautomatch (developed by K Zhang, https://www.mrc-lmb.cam.ac.uk/kzhang/Gautomatch/) and RELION. All other steps were performed using RELION 3.1 (*Zivanov et al., 2018*).

For the dataset of nRC-LH complex extracted from cells grown under high illumination (180 µmol m$^{-2}$ s$^{-1}$), the templates for automatic particle picking were 2D class averages of manually picked 3,106 particles. In total, 1,625,156 particles were auto-picked from 2,836 micrographs. The picked particles were extracted at 4×4 binning and subjected to two rounds of 2D classification. Good 2D class averages in different orientations were selected to generate the initial model. A subset of 1,041,360 particles at the original pixel size were selected for 3D classification into three classes with the initial model as a reference, and then 372,029 good particles were refined into a 3.7 Å resolution electron density map. Finally, the resultant data refined by per-particle CTF refinement were subjected to 3D refinement and postprocessing to 2.8 Å resolution on the gold-standard FSC (Fourier shell correlation)=0.143 criterion. The image processing of nRC-LH complexes extracted from cells grown under medium (32 µmol m$^{-2}$ s$^{-1}$) and low (2 µmol m$^{-2}$ s$^{-1}$) illuminations were summarized in *Figure 1—figure supplement 3*.

For the dataset of dRC-LH complex, a total of 1,081,719 particles were automatically picked from 3,514 micrographs. The picked particles were extracted at 4×4 binning and subjected to three rounds of reference-free 2D classification, resulting in 191,821 particles being left and re-extracted into the original pixel size of 0.893 Å. After 3D classification with three classes of particles, a subset of 84,352 particles was selected for the final refinement and postprocessing. The resolution of the final map was 3.1 Å. The values of the angular distribution of particles from 3D refinement were visualized by ChimeraX (*Pettersen et al., 2021*). Local resolution was estimated with ResMap (*Kucukelbir et al., 2014*).

## Model building and refinement

The reported 4.1 Å resolution model of RC-LH complex from *R. castenholzii* (PDB ID: 5YQ7) (*Xin et al., 2018*) was fitted into the density map in ChimeraX. Based on the density map, the structural model of the nRC-LH complex, including the amino acids residues, cofactors, lipids, and the newly identified exterior keto-γ-carotene (KγC$_{ext}$ and KγC) molecules were manually built and adjusted in Coot (*Emsley and Cowtan, 2004*). Then, real-space refinement in PHENIX (*Adams et al., 2010*) was used for model refinement with intra-cofactor and protein-cofactor geometric constraints. The structure of the dRC-LH complex was also manually built using the refined model of nRC-LH as a reference in COOT (*Emsley and Cowtan, 2004*) and refined using the real-space refinement in PHENIX (*Adams et al., 2010*). The refinement and model statistics are listed in *Table 2*.

## Assignment of the subunit X, proteins Y and Z

The cryo-EM map of nRC-LH was used for automated model building in ModelAngelo, a program developed by Prof. Sjors Scheres (https://arxiv.org/abs/2210.00006v1). BLAST search of the deduced amino acid sequences of subunit X generated a hint with hypothetical protein KatS3mg058_1126 (GenBank: GIV99722.1) from *Roseiflexus* sp., which was denoted by metagenomic analyses of the uncultivated bacteria in Katase hot spring sediment (*Kato et al., 2022*). However, this polypeptide has not been annotated in the Protein Database of *R. castenholzii* (strain DSM 13941/HLO8). By searching the genomic DNA of *R. castenholzii* (strain DSM 13941/HLO8), we eventually identified the coding sequences (CDS: 1,060,366–1,060,464) of subunit X, which shared strictly conserved amino acid sequence with KatS3mg058_1126. The assigned amino acid residues fitted well with the cryo-EM densities as shown in *Figure 2H*. Assignment of protein Y and Z was performed in same procedure, except that protein Z was also confirmed by PMF analyses shown in *Table 1*.

## Steady-state and fluorescence spectroscopy

Absorption spectra of the RC-LH complexes were collected at wavelength ranging from 250 to 900 nm using a UV-vis spectrophotometer (Mapada P6, Shanghai). Fluorescence emission and excitation spectra of the nRC-LH and dRC-LH complexes were recorded using a steady-state and time-resolved photoluminescence spectrometer (Edinburgh FLS1000, UK), equipped with a Hamamatsu NIR PMT detector (Hamamatsu Photonics, Japan) and an external adjustable 980 nm continuous-wave laser. The fluorescence excitation spectra were obtained with emissions monitored at 920 nm, and excitation at 470 nm was used for emission spectra.

## Ac oxidation assays

Isolation and purification of endogenous Ac from *R. castenholzii* was carried out by the methods as described (*Wang et al., 2020*). Before the oxidation assay, the purified Ac was treated with sodium dithionite to obtain the reduced Ac. Using the reduced Ac (122 µM) as electron donor and varied concentrations of menaquinone-4 (Sigma-Aldrich, USA) as electron acceptor, the reaction was carried out in the presence of nRC-LH or dRC-LH complex (50 nM) in buffer B (0.04% β-DDM, 100 mM NaCl, 50 mM Tris-HCl, pH 8.0). The reaction was initiated by illumination at 180 µmol m$^{-2}$ s$^{-1}$, and the absorbance of Ac at 604 nm was recorded by a UV-vis spectrophotometer (Mapada P6, Shanghai) at 2 min intervals for a total of 14 min. The corresponding concentrations of Ac were calculated with extinction coefficient, and linear initial rates from 2 to 14 min were fitted using the Michaelis-Menten model in Prism8. All data were obtained from three replicative experiments, with the mean and standard deviations calculated and plotted.

## Acknowledgements

We thank Prof. Fei Sun at the Institute of Biophysics, Chinese Academy of Science, and Prof. Weimin Ma at Shanghai Normal University for helpful discussions. We thank Danyu Gu from the Instrumentation and Service Center for Molecular Sciences at Westlake University for the assistance in measurement and data interpretation of the steady-state spectroscopic analyses. We appreciate the help from Prof. Kezhi Jiang of Hangzhou Normal University for the HPLC analysis of pigments. We thank the staff members of the Electron Microscopy System at the National Facility for Protein Science in Shanghai (NFPS), Zhangjiang Lab, China, for providing technical support and assistance in data collection of the dRC-LH complex. We also thank Shuimu BioSciences Ltd. for the support of cryo-EM data collection for the nRC-LH complex. Funding: This work was supported by grants from the National Natural Science Foundation of China (32171227, 31870740, 31570738 to XLX, 32301056 to JYX), Zhejiang Provincial Natural Science Foundation of China under Grant No. LR22C020002 to XLX and Zhejiang Provincial Education Department under Grant No. Y202044875 to JYX.

## Additional information

### Funding

| Funder | Grant reference number | Author |
|---|---|---|
| National Natural Science Foundation of China | 32171227 | Xiaoling Xu |
| National Natural Science Foundation of China | 31870740 | Xiaoling Xu |
| National Natural Science Foundation of China | 31570738 | Xiaoling Xu |
| Zhejiang Provincial Outstanding Youth Science Foundation | LR22C020002 | Xiaoling Xu |
| National Natural Science Foundation of China | 32301056 | Jiyu Xin |

The funders had no role in study design, data collection and interpretation, or the decision to submit the work for publication.

### Author contributions

Jiyu Xin, Data curation, Formal analysis, Funding acquisition, Validation, Investigation, Visualization, Methodology, Writing – original draft, Purified the RC-LH complexes, Determined the cryo-EM structures, Performed the enzymatic and steady-state spectroscopic analyses, Analyzed the data, Assisted with preparing the manuscript; Yang Shi, Data curation, Formal analysis, Validation, Investigation, Assigned the subunit X, protein Y and Z in the RC-LH complex; Xin Zhang, Data curation, Formal analysis, Validation, Investigation, Visualization, Assisted with the cryo-EM sample preparation and data processing, Analyzed the structures; Xinyi Yuan, Data curation, Formal analysis, Investigation, Extracted the pigments, Performed the HPLC-MS analyses, Analyzed the data; Yueyong Xin, Resources, Data curation, Investigation, Assisted with the spectral and HPLC-MS experiments; Huimin He, Data curation, Investigation, Assisted with the sample prepation and MS analysis; Jiejie Shen, Data curation, Formal analysis, Assisted with the auracyanin oxidation experiments; Robert E Blankenship, Writing – review and editing, Assisted with preparing the manuscript; Xiaoling Xu, Conceptualization, Resources, Formal analysis, Supervision, Funding acquisition, Validation, Investigation, Methodology, Writing – original draft, Project administration, Writing – review and editing, Initiated the project, Supervised all experiments, Analyzed the data, Wrote the manuscript

### Author ORCIDs

Jiyu Xin (ID) https://orcid.org/0000-0003-2354-2032
Xin Zhang (ID) https://orcid.org/0000-0002-6734-759X
Xiaoling Xu (ID) https://orcid.org/0000-0001-8995-1213

### Decision letter and Author response

Decision letter https://doi.org/10.7554/eLife.88951.sa1
Author response https://doi.org/10.7554/eLife.88951.sa2

## Additional files

### Supplementary files

• MDAR checklist

### Data availability

Cryo-EM maps and atomic coordinates of the native RC-LH (nRC-LH) and carotenoid depleted RC-LH (dRC-LH) complexes extracted from Roseiflexus castenholzii cells grown under high illumination (180 µmol m$^{-2}$ s$^{-1}$) have been deposited into the Electron Microscopy Data Bank (accession codes, EMD-34838 and EMD-34839) and the Protein Data Bank (PDB) (accession codes, 8HJU and 8HJV), respectively. Cryo-EM maps and atomic coordinates of the nRC-LH complexes extracted from cells

grown under low (2 μmol m$^{-2}$ s$^{-1}$) and medium (32 μmol m$^{-2}$ s$^{-1}$) illuminations have been deposited into the Electron Microscopy Data Bank (accession codes, EMD-35988 and EMD-35989) and the Protein Data Bank (PDB) (accession codes, 8J5O and 8J5P), respectively. Cryo-EM maps and atomic coordinates of the four RC-LH complexes can also be accessed on Dryad (https://doi.org/10.5061/dryad. w6m905qv4).

The following datasets were generated:

| Author(s) | Year | Dataset title | Dataset URL | Database and Identifier |
|---|---|---|---|---|
| Xin J | 2023 | Cryo-EM maps and atomic coordinates of RC-LH complexes from *Roseiflexus castenholzii* | https://doi.org/ 10.5061/dryad. w6m905qv4 | Dryad Digital Repository, 10.5061/dryad.w6m905qv4 |
| Xin J, Xu X | 2023 | Cryo-EM map of native RC-LH complex from *Roseiflexus castenholzii* at 180 μmol m$^{-2}$ s$^{-1}$ | https://www.ebi.ac. uk/emdb/EMD-34838 | Electron Microscopy Data Bank, EMD-34838 |
| Xin J, Xu X | 2023 | Cryo-EM map of carotenoid-depleted RC-LH complex from *Roseiflexus castenholzii* at 180 μmol m$^{-2}$ s$^{-1}$ | https://www.ebi.ac. uk/emdb/EMD-34839 | Electron Microscopy Data Bank, EMD-34839 |
| Xin J, Xu X | 2023 | Cryo-EM structure of native RC-LH complex from *Roseiflexus castenholzii* at 180 μmol m$^{-2}$ s$^{-1}$ | https://www.rcsb.org/ structure/8HJU | RCSB Protein Data Bank, 8HJU |
| Xin J, Xu X | 2023 | Cryo-EM structure of carotenoid-depleted RC-LH complex from *Roseiflexus castenholzii* at 180 μmol m$^{-2}$ s$^{-1}$ | https://www.rcsb.org/ structure/8HJV | RCSB Protein Data Bank, 8HJV |
| Xin J, Xu X | 2023 | Cryo-EM map of native RC-LH complex from *Roseiflexus castenholzii* at 2 μmol m$^{-2}$ s$^{-1}$ | https://www.ebi.ac. uk/emdb/EMD-35988 | Electron Microscopy Data Bank, EMD-35988 |
| Xin J, Xu X | 2023 | Cryo-EM map of native RC-LH complex from *Roseiflexus castenholzii* at 32 μmol m$^{-2}$ s$^{-1}$ | https://www.ebi.ac. uk/emdb/EMD-35989 | Electron Microscopy Data Bank, EMD-35989 |
| Xin J, Xu X | 2023 | Cryo-EM structure of native RC-LH complex from *Roseiflexus castenholzii* at 2 μmol m$^{-2}$ s$^{-1}$ | https://www.rcsb.org/ structure/8J5O | RCSB Protein Data Bank, 8J5O |
| Xin J, Xu X | 2023 | Cryo-EM structure of native RC-LH complex from *Roseiflexus castenholzii* at 32 μmol m$^{-2}$ s$^{-1}$ | https://www.rcsb.org/ structure/8J5P | RCSB Protein Data Bank, 8J5P |

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
