## [Editor Report]

This is a valuable analysis of the structure of *Roseiflexus castenholzii* native and carotenoid-depleted light harvesting complexes. The authors have investigated the relationship between Carotenoid pigment depletion in the photosynthesis-related light harvesting complex, the assembly of the prokaryotic reaction center LH complex, and quinone exchange in *Roseiflexus castenholzii*, a chlorosome-less filamentous anoxygenic phototroph that forms the deepest branch of photosynthetic bacteria. The evidence supporting the claims is solid, with application of rigorous biochemical and biophysical techniques, including cryo-electron microscopy of the purified RC-LH complexes with or depleted of carotenoids. This study will be of interest to biologists working on the evolution and diversity of prokaryotic photosynthetic apparatus.

---

## [Decision Letter]

**Decision letter after peer review:**

Thank you for submitting your article "Carotenoid assembly regulates quinone diffusion and the *Roseiflexus castenholzii* reaction center-light harvesting complex architecture" for consideration by *eLife*. Your article has been reviewed by 3 peer reviewers, one of whom is a member of our Board of Reviewing Editors, and the evaluation has been overseen by Amy Andreotti as the Senior Editor.

Essential revisions:

1) The authors included in the manuscript some predictions about the functionality of some of the results obtained by structural analysis of the native and carotenoid-depleted LHs. However, because these are predictions, some of the statements should be toned down.

2) In the dRC-LH structure, the authors observe five internal carotenoid molecules bound to LH5, 7, 9, 10 and 11, respectively. However, why these five irregular positions can stably accommodate carotenoids is a bit difficult to understand. The authors need to clarify whether they built only five carotenoids in the structure, or the complex contains only five carotenoid molecules.

3) When the effect of growth at high light was analyzed, the high light structures were taken after very long exposure. Thus, these structures seem to represent partially assembled or partially destroyed complexes. This possibility should be at least discussed.

*Reviewer #1 (Recommendations for the authors):*

1. As I mentioned in the Public Review, it is interesting that the authors included in the manuscript some predictions about the functionality of some of the results obtained by structural analysis of the native and carotenoid-depleted LHs. However, because these are predictions, I think that some of the statements should be toned down. For example, when the role of the gap in the ring that persists without a canonical subunit X was analyzed, it is concluded: " This conformation accelerated quinone/quinol exchange without affecting the Car-to-BChl energy transfer efficiency. " (lines 106-107). Although this seems to support the view that the X-less gap is functional, as the work compares very different species and uses a soluble menaquinone analog, which can diffuse differently than the native long-tail version, the authors should be more cautious about this conclusion.

2. When the effect of growth at high light was analyzed, the high light structures were taken after very long exposure. Thus, these structures seem to represent partially assembled or partially destroyed complexes. In my opinion, the results are difficult to interpret.

3. The use of DPA, as also mentioned by the authors (lines 469-471) is not a very clean way to examine the effect of carotenoids, as it left several interior carotenoids still bound to the LH ring. I think that it would be interesting to use a *Roseiflexus castenholzii* mutant in an enzyme of the carotenoid pathway, for example phytoene desaturase. I am not sure of how easy it is to genetically manipulate this bacteria, but the structural and in vivo (photosynthesic measurements, cell proliferation assays, etc.) analysis of completely carotenoid depleted LH would be very valuable to validate the results presented.

*Reviewer #2 (Recommendations for the authors):*

In the dRC-LH structure, the authors observe five internal carotenoid molecules bound to LH5, 7, 9, 10 and 11, respectively. However, why these five irregular positions can stably accommodate carotenoids is a bit difficult to understand. The authors need to clarify whether they built only five carotenoids in the structure, or the complex contains only five carotenoid molecules. Please specify if only these five carotenoid molecules show sharp density while other equivalent binding sites (particularly LH6 and LH8) do not? If so, the authors are encouraged to compare the structures of LH dimers composed of two adjacent LHs (5-6, 6-7, 7-8, etc.) to see if there are structural differences between different dimeric LHs.

The authors find that dRC-LH is increased in size compared to nRC-LH. This statement should be made with extreme caution if the two datasets were collected on different cryo-EM facilities. The dimension of the reconstruction map is related to the value of pixel size used for data processing, thus slight inaccuracy in pixel size may cause the increased/decreased sizes of the reconstruction map.

L209, L438: I do not think that "*R. castenholzii* RC-LH has evolved different quinone shuttling mechanisms". It may be that *R. castenholzii* RC-LH has evolved different structural elements to regulate quinone shuttling, but the mechanism is similar to others.

L92: Please use the standard unit of light intensity, umol photon m(-2) s(-1) instead of lux.

Table 2: The values of clash score are quite high compared to other structures with similar resolution. The authors could try to improve the quality by including hydrogens for refinement.

*Reviewer #3 (Recommendations for the authors):*The article has many figures and supplementary figures to support the results. It will be great if authors could indicate in each which is the cytoplasmatic or periplasmic site, maybe with a letter just to guide the reader (example: line 154 figure 2 and SF1A).

In the paper, every time the authors should mention *Roseiflexus castenholzii* after they already mention one they should use: *R. castenholzii*. As well in some parts is not "italic" written.

The references should be reviewed, for example, some of them have several authors mentioned in the text, for example: Xin, Pan, Collins, Lin, and Blankenship, 2012, line 321.

In some parts, such as line 224, the large amount of information provided by the detailed figures makes the reader get lost. For example, this line where there is a Met25 and a Val25 which are from different proteins. Maybe the authors should use a line or highlight colour or a way to focus the residues. It is just a suggestion, I understood that it is deeply described and in one way it is difficult to direct the focus to each residue and more in this case that there are several proteins and several similar numberings.

---

## [Author Response]

Essential revisions:Reviewer #1 (Recommendations for the authors):1. As I mentioned in the Public Review, it is interesting that the authors included in the manuscript some predictions about the functionality of some of the results obtained by structural analysis of the native and carotenoid-depleted LHs. However, because these are predictions, I think that some of the statements should be toned down. For example, when the role of the gap in the ring that persists without a canonical subunit X was analyzed, it is concluded: " This conformation accelerated quinone/quinol exchange without affecting the Car-to-BChl energy transfer efficiency. " (lines 106-107). Although this seems to support the view that the X-less gap is functional, as the work compares very different species and uses a soluble menaquinone analog, which can diffuse differently than the native long-tail version, the authors should be more cautious about this conclusion.

Thanks for the valuable comments, we agree with the reviewer and have toned down the statements about the functionality of some structural observations.

The predominant quinone of *R. castenholzii* isolates was determined to be menaquinone (MK)-11 (Hanada, S. et al., *Int J Syst Evol Microbiol* 2002, 52:187-193.), which contains a longer isoprenoid tail and less solubility than menaquinone-4 (MK-4) that we used in the auracyanin (Ac) oxidation assay. In our reaction system, the maximum solubility of MK-4 was determined to be approximately 80 μM. We agree with the reviewer that the native MK-11 with a long isoprenoid tail would exhibit less solubility and different diffuse rate as compared to MK-4.

To avoid overstatement of the role of LH ring opening, we have toned down the statement to:

“This conformation accelerated the in vitro quinone/quinol exchange rate of menaquinone-4, an analog of the native menaquinone-11, but did not affect the Car-to-BChl energy transfer efficiency of dRC-LH.” at lines 104-106, 411 and 493.

2. When the effect of growth at high light was analyzed, the high light structures were taken after very long exposure. Thus, these structures seem to represent partially assembled or partially destroyed complexes. In my opinion, the results are difficult to interpret.

Thanks for the valuable comments. Our results showed that light intensity affected the proliferation rate of *R. castenholzii*. The cells cultured under high light intensity showed much faster proliferation rate than that grown under medium and low illuminations (Figure 1—figure supplement 1B). However, light intensities did not affect the structures (including the protein subunits, pigments and cofactors) of three native RC-LH complexes extracted from *R. castenholzii* cells grown under high (180 μmol m^-2^ s^-1^), medium (32 μmol m^-2^ s^-1^) and low (2 μmol m^-2^ s^-1^) illuminations, respectively. And the high light structure does not represent partially assembled or partially destroyed complex, since it contains the complete protein subunits, pigments and cofactors as the medium and low light structures (Figure 1; Figure 1—figure supplement 5).

First, the same amount of native RC-LH complexes extracted from cells grown under high, medium and low light intensities showed no difference in the absorption spectrum (Figure 1—figure supplement 1E), indicating the light intensity did not affect the pigments composition and content in the native RC-LH complex. (see lines 126-129).

Second, the native RC-LH structures obtained at high, medium, and low illuminations showed the same subunit composition and architecture, as well as the configurations of the assembled pigments and cofactors (Figure 1—figure supplement 5). To highlight the similarity between protein structures, we have included the statement at lines 133-136:

“Superposition of the high illumination model with that of medium and low illumination gave root mean square deviation (RMSD) of 1.753 Å and 1.765 Å, respectively, indicating these three structures share the same architecture, and light intensities did not affect the conformation of the nRC-LH structures.” We have also described the similarity between Car organizations at lines 144-148: “In particular, the LH ring bound 15 interior (KγC_int_), 14 exterior (KγC_ext_) Cars, and an additional KγC that inserted between the LHαβ1 and c-TM in all three structures (Figure 1B-D, Figure 1—figure supplement 5, Video 1), indicating both Car compositions and assembly in the nRC-LH were not affected by light intensities.”

Third, the high light structure contains the complete protein subunits, pigments and cofactors (Figure 1; Figure 1—figure supplement 5). Therefore, the high light structure does not represent partially assembled or partially destroyed complex. These observations indicated that the light intensity does not affect the structure, and assembly of Cars and BChls into the native RC-LH complex.

3. The use of DPA, as also mentioned by the authors (lines 469-471) is not a very clean way to examine the effect of carotenoids, as it left several interior carotenoids still bound to the LH ring. I think that it would be interesting to use a Roseiflexus castenholzii mutant in an enzyme of the carotenoid pathway, for example phytoene desaturase. I am not sure of how easy it is to genetically manipulate this bacteria, but the structural and in vivo (photosynthesic measurements, cell proliferation assays, etc.) analysis of completely carotenoid depleted LH would be very valuable to validate the results presented.

Thanks for the valuable comments. We agree with the reviewers that an easy-to-handle genetic manipulation system of *Roseiflexus castenholzii* is necessary for investigating the effects of carotenoids. However, we met some difficulties in constructing a genetic manipulation system of *R. castenholzii* (as following), and we also mentioned these limitations at lines 485-488.

First, *R. castenholzii* is a multicellular filamentous anoxygenic bacterium, it exhibits a filamentous morphology instead of forming single colonies on the solid culture medium (Hanada, S. et al., *Int J Syst Evol Microbiol* 2002, 52: 187-193). This morphology brings on complexity and challenges in selection and isolation of the single recombinant. In addition, the optimal growth temperature of *R. castenholzii* is around 50 °C, which causes antibiotic deactivation and reduced efficiency in resistance-based selection. While the well-established model bacteria often contain an optimal growth temperature at 25~35 °C, which does not affect the efficiency of the inserted resistance marker and the conjugation/transduction operations.

Second, *R. castenholzii* does not contain a well-studied genetic background. Only few of the well-studied purple non-sulfur photosynthetic bacteria, like *Rhodopseudomonas palustris* and *Rhodobacter capsulatus*, have developed efficient genetic manipulation systems. These photosynthetic bacteria contain well-established genetic background that facilitates the exogenous DNA introduction and replication, including high-efficient DNA introduction (conjugation/transduction) methods, well-defined endogenous promoters and restriction-modification systems.

However, the genetic regulation mechanisms of *R. castenholzii* have not been thoroughly studied. For example, a plasmid shuttle vector containing a controllable gene induction system is required for genetic manipulation. The shuttle vector of thermophilic bacteria is usually derived from their natural plasmids. Such as the hyperthermophilic archaeon *Sulfolobus*, it contains a natural plasmid pRN1 that could be modified to obtain a shuttle plasmid for exogenous gene introduction (Berkner et al., *Nucleic Acids Res* 2007, 35(12): e88). In contrast, *R. castenholzii* itself does not possess natural plasmids, and no shuttle plasmids between *E. coli* and *R. castenholzii* has been reported. Therefore, it will take a very long time to screen and test various shuttle vectors, and to explore and understand the genetic regulation system of *R. castenholzii* for efficient gene editing.

Actually, we have also tried to introduce suicide plasmids and use homologous recombination to import exogenous resistance genes into *R. castenholzii*. However, it was very difficult to select the positive strains that carrying resistance genes, due to above morphological and culturing properties of *R. castenholzii*, and also the low plasmid introduction efficiency. Overall, these characteristics of *R. castenholzii* bring on difficulties in constructing an effective genetic manipulation system in a relatively short time. Due to lack of experience on genetic manipulations of thermophilic bacteria, we will collaborate with experts on this field to solve these problems. Nevertheless, constructing an efficient genetic manipulation system will be the subject of future work but is beyond the scope of the current paper.

Reviewer #2 (Recommendations for the authors):In the dRC-LH structure, the authors observe five internal carotenoid molecules bound to LH5, 7, 9, 10 and 11, respectively. However, why these five irregular positions can stably accommodate carotenoids is a bit difficult to understand. The authors need to clarify whether they built only five carotenoids in the structure, or the complex contains only five carotenoid molecules. Please specify if only these five carotenoid molecules show sharp density while other equivalent binding sites (particularly LH6 and LH8) do not? If so, the authors are encouraged to compare the structures of LH dimers composed of two adjacent LHs (5-6, 6-7, 7-8, etc.) to see if there are structural differences between different dimeric LHs.

Thanks for the valuable comments. To avoid confusion, we have clarified the building of five interior carotenoid (KγC_int_) molecules in dRC-LH at lines 302-303, and the effects of carotenoids binding on LHαβ structure at lines 313-320.

High performance liquid chromatography (HPLC) analyses of the pigments extracted from dRC-LH complex showed the same pigments composition but strikingly decreased carotenoids absorbance compared to the nRC-LH (Figure 2—figure supplement 3A and B). Mass spectrometry (MS) analysis of these HPLC peaks further revealed that both nRC-LH and dRC-LH contain γ-carotene and several γ-carotene derivatives (Figure 2—figure supplement 3C and D). However, the content of the carotenoids was unable to be accurately quantified, due to lack of standard samples of these endogenously synthesized γ-carotene derivatives and their specific absorption coefficients.

If we use the molar extinction coefficient of γ-carotene (ɛ_475_ = 167 mM^-1^ cm^-1^) and BChls (ɛ_770_ = 60 mM^-1^ cm^-1^) to roughly integrate the corresponding HPLC peaks, we can obtain a Car: BChl ratio of approximately 1:1.94 for the nRC-LH and 1:13.97 for the dRC-LH, respectively. The roughly calculated Car: BChl ratio in nRC-LH (1:1.94) is close to but not exactly match the built Car: BChl molecules of 30:48 (1:1.6). However, the built Cars and BChls numbers are consistent with previous pigment analyses, which have revealed a Car: BChl ratio of 2:3 for *R. castenholzii* RC-LH (Collins et al., *Biochim Biophys Acta* 2009, 1787(8): 1050-1056.).

For dRC-LH, although we revealed that DPA treatment indeed decreased the content of LH-bound carotenoids, but the exact quantity of carotenoids was also unable to be accurately calculated. In the cryo-EM map, only the five KγC_int_ molecules bound to LHαβ5, 7, 9, 10, and 11 showed sharp densities, while the densities at other equivalent binding sites were not clear (Revised Figure 4—figure supplement 2A). Therefore, based on the visible cryo-EM densities and decreased carotenoids content, we built five KγC_int_ molecules in the dRC-LH structure. This resulted in a built Car: BChl ratio of 5:48 (1:9.6) in the dRC-LH structure, which is even larger than the roughly calculated Car: BChl ratio (1:13.97) of the dRC-LH sample. Specifically, both the LHαβ dimers and bound carotenoids showed high B-factors near the LH opening (Figure 2—figure supplement 2), indicating the carotenoids near the LH opening are less stable and easier to be dissociated from the LH dimer. It is understandable that the five KγC_int_ molecules are located relatively far from the LH opening (~52 Å), which is where Cars with the highest B-factors are distributed, indicating an unstable conformation (Figure 2—figure supplement 2A). Thus, building of these five KγC_int_ molecules most likely a true representation of the dramatically decreased carotenoids content in the dRC-LH.

Superposition of the LHαβ5 heterodimer with LHαβ6, 7, and 8 gave a root-mean-square deviation (RMSD) of 0.001 Å, indicating that the structures of these adjacent LHαβs are essentially identical (Revised Figure 4—figure supplement 2B). To be noted, we observed slight differences at the LHα-Phe28 sidechain orientations. In the dRC-LH structure, the Phe28 sidechains in LHα7, 9, and 11 were oriented toward the RC, while the others were oriented towards the adjacent LHα subunits (Revised Figure 4—figure supplement 2C). Similar sidechain orientations of Phe28 were observed in the nRC-LH structure at LHα7, 9, 11, and LHα15 that binds both KγC_int_ and exterior carotenoid (KγC_ext_) molecules (Revised Figure 4—figure supplement 2D). Since each of the 15 LHαβs in nRC-LH binds both the KγC_int_ and KγC_ext_, these observations indicate that the sidechain orientations of Phe28 are not correlated with the carotenoid binding. In addition, no direct interactions were observed between the LH dimer (including Phe28) and bound carotenoids (Figure 5A and B). Thus, the LH dimer structures are not affected by carotenoids depletion in the dRC-LH.

Overall, DPA treatment inhibited carotenoids incorporation into the dRC-LH complex. The five KγC_int_ molecules resolved at LHαβ5, 7, 9, 10, and 11 of the dRC-LH were built according to the dramatically decreased carotenoid content and clearly visible cryo-EM densities. The dRC-LH contains much less carotenoids that could not be accurately quantified, due to the complicated composition and lack of specific absorption coefficients of the γ-carotene derivatives.

The authors find that dRC-LH is increased in size compared to nRC-LH. This statement should be made with extreme caution if the two datasets were collected on different cryo-EM facilities. The dimension of the reconstruction map is related to the value of pixel size used for data processing, thus slight inaccuracy in pixel size may cause the increased/decreased sizes of the reconstruction map.

Thanks for the valuable comments. We agree with the reviewers that variations in data collection parameters may affect the dimensions of structures when using different cryo-EM facilities. Therefore, we have removed the description of the changes in LH ring size.

L209, L438: I do not think that "R. castenholzii RC-LH has evolved different quinone shuttling mechanisms". It may be that R. castenholzii RC-LH has evolved different structural elements to regulate quinone shuttling, but the mechanism is similar to others.

We appreciate the valuable comments. We agree with the reviewers and have changed the statements at lines 212 and 446 in the revised manuscript.

L92: Please use the standard unit of light intensity, umol photon m(-2) s(-1) instead of lux.

Thanks for the valuable comments. We have changed the unit of light intensity from lux to μmol m^-2^ s^-1^ in the revised manuscript. For easier reading, we labeled the light intensities at 10,000, 2,000 and 100 lux as high (180 μmol m^-2^ s^-1^), medium (32 μmol m^-2^ s^-1^), and low (2 μmol m^-2^ s^-1^) illuminations, respectively.

Table 2: The values of clash score are quite high compared to other structures with similar resolution. The authors could try to improve the quality by including hydrogens for refinement.

Thanks for the valuable comments. After including hydrogens for the refinement, we decreased clash score of the nRC-LH structural model to 16.11 (Revised Table 2).

Reviewer #3 (Recommendations for the authors):The article has many figures and supplementary figures to support the results. It will be great if authors could indicate in each which is the cytoplasmatic or periplasmic site, maybe with a letter just to guide the reader (example: line 154 figure 2 and SF1A).

Thanks for the valuable comments. We have labeled the periplasmic (P) or cytoplasmic (C) sides in each related figure for easier reading.

In the paper, every time the authors should mention Roseiflexus castenholzii after they already mention one they should use: R. castenholzii. As well in some parts is not "italic" written.

Thanks for the valuable comments. We have modified “*Roseiflexus castenholzii*” to “*R. castenholzii*” after first time mention it, and changed the fonts in italics.

The references should be reviewed, for example, some of them have several authors mentioned in the text, for example: Xin, Pan, Collins, Lin, and Blankenship, 2012, line 321.

Thanks for the valuable comments. We have corrected the pattern of citing references.

In some parts, such as line 224, the large amount of information provided by the detailed figures makes the reader get lost. For example, this line where there is a Met25 and a Val25 which are from different proteins. Maybe the authors should use a line or highlight colour or a way to focus the residues. It is just a suggestion, I understood that it is deeply described and in one way it is difficult to direct the focus to each residue and more in this case that there are several proteins and several similar numberings.

Thanks for the valuable comments. We have included the subunit information for the key residues in the revised Figure 2I and other figures depicting residues.